# Pseudo-Spherical Contrastive Divergence

**Lantao Yu**
Computer Science Department
Stanford University
`lantaoyu@cs.stanford.edu`

**Jiaming Song**
Computer Science Department
Stanford University
`tsong@cs.stanford.edu`

**Yang Song**
Computer Science Department
Stanford University
`yangsong@cs.stanford.edu`

**Stefano Ermon**
Computer Science Department
Stanford University
`ermon@cs.stanford.edu`

## Abstract

Energy-based models (EBMs) offer flexible distribution parametrization. However, due to the intractable partition function, they are typically trained via contrastive divergence for maximum likelihood estimation. In this paper, we propose pseudo-spherical contrastive divergence (PS-CD) to generalize maximum likelihood learning of EBMs. PS-CD is derived from the maximization of a family of strictly proper homogeneous scoring rules, which avoids the computation of the intractable partition function and provides a generalized family of learning objectives that include contrastive divergence as a special case. Moreover, PS-CD allows us to flexibly choose various learning objectives to train EBMs without additional computational cost or variational minimax optimization. Theoretical analysis on the proposed method and extensive experiments on both synthetic data and commonly used image datasets demonstrate the effectiveness and modeling flexibility of PS-CD, as well as its robustness to data contamination, thus showing its superiority over maximum likelihood and $f$-EBMs.

## 1 Introduction

Energy-based models (EBMs) provide a unified framework for generative and discriminative learning by capturing dependencies between random variables with an energy function. Due to the absence of the normalization constraint, EBMs offer much more flexibility in distribution parametrization and architecture design compared to properly normalized probabilistic models such as autoregressive models [53, 23], flow-based models [14, 15, 46] and sum-product networks [72]. Recently, deep EBMs have achieved considerable success in realistic image generation [17, 66, 12, 30], molecular modeling [90] and model-based planning [16], thanks to modern deep neural networks [54, 49, 35] for parametrizing expressive energy functions and improved Markov Chain Monte Carlo (MCMC) techniques [62, 75, 40, 17, 66] for efficiently sampling from EBMs.

Training EBMs consists of finding an energy function that assigns low energies to correct configurations of variables and high energies to incorrect ones [55], where a central concept is the *loss functional* that is used to measure the quality of the energy function and is minimized during training. The flexibility of EBMs does not come for free: it makes the design of loss functionals particularly challenging, as it usually involves the partition function that is generally intractable to compute. As a result, EBMs are typically trained via CD [37], which belongs to the "analysis by synthesis" scheme [31] and performs a sampling-based estimation of the gradient of KL between data distribution and energy-based distribution. Since different loss functionals will induce different solutions in practice (when the model is mis-specified and data is finite) and KL may not provide the right inductive

bias [25, 91], inspired by the great success of implicit generative models [28, 67, 3], [89] proposed a variational framework to train EBMs by minimizing general $f$-divergences [10]. Although this framework enables us to specify various modeling preferences such as diversity/quality tradeoff, they rely on learning additional components (variational functions) within a minimax framework, where the optimization is complicated by the notion of Nash equilibrium and local optimality [42] and suffers from instability and non-convergence issues [59]. Along this line, noise contrastive estimation (NCE) [34] can train EBMs with a family of loss functionals induced by different Bregman divergences. However, in practice, NCE usually relies on carefully-designed noise distribution such as context-dependent noise distribution [41] or joint learning of a flow-based noise distribution [21].

In this paper, we draw inspiration from statistical decision theory [13] and propose a novel perspective for designing loss functionals for training EBMs without involving auxiliary models or variational optimization. Specifically, to generalize maximum likelihood training of EBMs, we focus on maximizing pseudo-spherical scoring rules [76, 27], which are *strictly proper* such that the data distribution is the unique optimum and *homogeneous* such that they can be evaluated without the knowledge of the normalization constant. Under the "analysis by synthesis" scheme used in CD and $f$-EBM [89], we then derive a practical algorithm termed Pseudo-Spherical Contrastive Divergence (PS-CD). Different from $f$-EBM, PS-CD enables us to specify flexible modeling preferences without requiring additional computational cost or unstable minimax optimization. We provide a theoretical analysis on the sample complexity and convergence property of PS-CD, as well as its connections to maximum likelihood training. With experiments on both synthetic data and commonly used image datasets, we show that PS-CD achieves significant sample quality improvement over conventional maximum likelihood training and competitive performance to $f$-EBM without expensive variational optimization. Based on a set of recently proposed generative model evaluation metrics [61], we further demonstrate the various modeling tradeoffs enabled by PS-CD, justifying its modeling flexibility. Moreover, PS-CD is also much more robust than CD in face of data contamination.

## 2 Preliminaries

### 2.1 Energy-Based Distribution Representation and Sampling

Given a set of *i.i.d.* samples $\{\boldsymbol{x}_i\}_{i=1}^{N}$ from some unknown data distribution $p(\boldsymbol{x})$ defined over the sample space $\mathcal{X} \subset \mathbb{R}^m$, the goal of generative modeling is to learn a $\boldsymbol{\theta}$-parametrized probability distribution $q_{\boldsymbol{\theta}}(\boldsymbol{x})$ to approximate the data distribution $p(\boldsymbol{x})$. In the context of energy-based modeling, instead of directly parametrizing a properly normalized distribution, we first parametrize an unnormalized energy function $E_{\boldsymbol{\theta}} : \mathcal{X} \to \mathbb{R}$, which further defines a normalized probability density via the Boltzmann distribution:

$$q_{\boldsymbol{\theta}}(\boldsymbol{x}) = \frac{\overline{q}_{\boldsymbol{\theta}}(\boldsymbol{x})}{Z_{\boldsymbol{\theta}}} = \frac{\exp(-E_{\boldsymbol{\theta}}(\boldsymbol{x}))}{Z_{\boldsymbol{\theta}}}, \tag{1}$$

where $Z_{\boldsymbol{\theta}} := \int_{\mathcal{X}} \exp(-E_{\boldsymbol{\theta}}(\boldsymbol{x}))\mathrm{d}\boldsymbol{x}$ is the partition function (normalization constant). In this paper, unless otherwise stated, we will use $\overline{q}$ to denote an unnormalized density and $q$ to denote the corresponding normalized distribution. We also assume that the exponential of the negative energy belongs to the $L^1$ space, $\mathcal{E} := \{E_{\boldsymbol{\theta}} : \mathcal{X} \to \mathbb{R} : \int_{\mathcal{X}} \exp(-E_{\boldsymbol{\theta}}(\boldsymbol{x}))\mathrm{d}\boldsymbol{x} < \infty\}$, *i.e.*, $Z_{\boldsymbol{\theta}}$ is finite.

Since energy-based models (EBMs) represent a probability distribution by assigning unnormalized scalar values (energies) to the data points, we can use any model architecture that outputs a bounded real number given an input to implement the energy function, which allows extreme flexibility in distribution parametrization. However, it is non-trivial to sample from an EBM, usually requiring MCMC [75] techniques. Specifically, in this work we consider using Langevin dynamics [62, 88], a gradient-based MCMC method that performs noisy gradient descent to traverse the energy landscape and arrive at the low-energy configurations:

$$\tilde{\boldsymbol{x}}_t = \tilde{\boldsymbol{x}}_{t-1} - \frac{\epsilon}{2}\nabla_{\boldsymbol{x}}E_{\boldsymbol{\theta}}(\tilde{\boldsymbol{x}}_{t-1}) + \sqrt{\epsilon}\boldsymbol{z}_t, \tag{2}$$

where $\boldsymbol{z}_t \sim \mathcal{N}(0, I)$. The distribution of $\tilde{\boldsymbol{x}}_T$ converges to the model distribution $q_{\boldsymbol{\theta}}(\boldsymbol{x}) \propto \exp(-E_{\boldsymbol{\theta}}(\boldsymbol{x}))$ when $\epsilon \to 0$ and $T \to \infty$ under some regularity conditions [88]. In order to sample from an energy-based distribution efficiently, many scalable techniques have been proposed such as learning non-convergent, non-persistent, short-run MCMC [66] and using a sample replay buffer to improve mixing time and sample diversity [17]. In this work, we leverage these recent advances when we need to obtain samples from an EBM.

## 2.2 Maximum Likelihood Training of EBMs via Contrastive Divergence

The predominant approach to training explicit density generative models is to approximately minimize the KL divergence between the (empirical) data distribution and model distribution. Minimizing KL divergence is equivalent to the following maximum likelihood estimation (MLE) objective:

$$\min_{\boldsymbol{\theta}} \mathcal{L}_{\mathrm{MLE}}(\boldsymbol{\theta}; p) = \min_{\boldsymbol{\theta}} -\mathbb{E}_{p(\boldsymbol{x})}[\log q_{\boldsymbol{\theta}}(\boldsymbol{x})] = \min_{\boldsymbol{\theta}} \mathbb{E}_{p(\boldsymbol{x})}[E_{\boldsymbol{\theta}}(\boldsymbol{x})] + \log Z_{\boldsymbol{\theta}}. \tag{3}$$

Because of the intractable partition function (an integral over the sample space), we cannot directly optimize the above MLE objective. To tackle this issue, [37] proposed contrastive divergence (CD) algorithm as a convenient way to estimate the gradient of $\mathcal{L}_{\mathrm{MLE}}(\boldsymbol{\theta}; p)$ using samples from $q_{\boldsymbol{\theta}}$:

$$\nabla_{\boldsymbol{\theta}} \mathcal{L}_{\mathrm{MLE}}(\boldsymbol{\theta}; p) = \mathbb{E}_{p(\boldsymbol{x})}[\nabla_{\boldsymbol{\theta}} E_{\boldsymbol{\theta}}(\boldsymbol{x})] + \nabla_{\boldsymbol{\theta}} \log Z_{\boldsymbol{\theta}} = \mathbb{E}_{p(\boldsymbol{x})}[\nabla_{\boldsymbol{\theta}} E_{\boldsymbol{\theta}}(\boldsymbol{x})] - \mathbb{E}_{q_{\boldsymbol{\theta}}(\boldsymbol{x})}[\nabla_{\boldsymbol{\theta}} E_{\boldsymbol{\theta}}(\boldsymbol{x})], \tag{4}$$

which can be interpreted as decreasing the energies of real data from $p$ and increasing the energies of fake data generated by $q_{\boldsymbol{\theta}}$. As discussed above, evaluating Equation (4) typically relies on MCMC methods such as the Langevin dynamics sampling procedure defined in Equation (2) to produce samples from the model distribution $q_{\boldsymbol{\theta}}$, which induces a surrogate gradient estimation:

$$\nabla_{\boldsymbol{\theta}} \mathcal{L}_{\mathrm{CD}-K}(\boldsymbol{\theta}; p) = \mathbb{E}_{p(\boldsymbol{x})}[\nabla_{\boldsymbol{\theta}} E_{\boldsymbol{\theta}}(\boldsymbol{x})] - \mathbb{E}_{q_{\boldsymbol{\theta}}^K(\boldsymbol{x})}[\nabla_{\boldsymbol{\theta}} E_{\boldsymbol{\theta}}(\boldsymbol{x})], \tag{5}$$

where $q_{\boldsymbol{\theta}}^K$ denotes the distribution after $K$ steps of MCMC transitions from an initial distribution (typically data distribution or uniform distribution), and Equation (4) corresponds to $\mathcal{L}_{\mathrm{CD}-\infty}$.

## 2.3 Strictly Proper Scoring Rules

Stemming from statistical decision theory [13], scoring rules evaluate the quality of probabilistic forecasts by assigning numerical scores based on the predictive distributions and the events that materialize. Formally, consider a compact sample space $\mathcal{X}$. Let $\mathcal{M}$ be a space of all locally 1-integrable non-negative finite measures and $\mathcal{P}$ be a subspace consisting of all probability measures on the sample space $\mathcal{X}$. A scoring rule $S(\boldsymbol{x}, q)$ specifies the *utility* of forecasting using a probability forecast $q \in \mathcal{P}$ for a given sample $\boldsymbol{x} \in \mathcal{X}$. With slightly abused notation, we write the *expected score* of $S(\boldsymbol{x}, q)$ under a reference distribution $p$ as:

$$S(p, q) := \mathbb{E}_{p(\boldsymbol{x})}[S(\boldsymbol{x}, q)]. \tag{6}$$

**Definition 1** (Proper Scoring Rules [26]). *A scoring rule $S : \mathcal{X} \times \mathcal{P} \to \mathbb{R}$ is called proper relative to $\mathcal{P}$ if the corresponding expected score satisfies:*

$$\forall p, q \in \mathcal{P}, S(p, q) \leq S(p, p). \tag{7}$$

*It is strictly proper if the equality holds if and only if $q = p$.*

In prediction and elicitation problems, strictly proper scoring rules encourage the forecaster to make honest predictions based on their true beliefs [22]. In estimation problems, where we want to approximate a distribution $p$ with another parametric distribution $q_{\boldsymbol{\theta}}$, strictly proper scoring rules provide attractive learning objectives:

$$\arg\max_{q_{\boldsymbol{\theta}} \in \mathcal{P}_{\boldsymbol{\theta}}} S(p, q_{\boldsymbol{\theta}}) = \arg\max_{q_{\boldsymbol{\theta}} \in \mathcal{P}_{\boldsymbol{\theta}}} \mathbb{E}_{p(\boldsymbol{x})}[S(\boldsymbol{x}, q_{\boldsymbol{\theta}})] = p \text{ (when } p \in \mathcal{P}_{\boldsymbol{\theta}}). \tag{8}$$

When a scoring rule $S$ is strictly proper relative to $\mathcal{P}$, the associated *generalized entropy function* and *divergence function* are defined as:

$$G(p) := \sup_{q \in \mathcal{P}} S(p, q) = S(p, p), \quad D(p, q) := S(p, p) - S(p, q) \geq 0. \tag{9}$$

$G(p)$ is convex and represents the maximally achievable utility, while $D(p, q)$ is the Bregman divergence [6] associated with the convex function $G$ and the equality holds only when $p = q$.

Next, we introduce a specific kind of scoring rules that are particularly suitable for learning unnormalized statistical models.

**Definition 2** (Homogeneous Scoring Rules [69]). *A scoring rule is homogeneous if it satisfies (here the domain of the score function is extended to $\mathcal{X} \times \mathcal{M}$):*

$$\forall \lambda > 0, \boldsymbol{x} \in \mathcal{X}, \quad S(\boldsymbol{x}, q) = S(\boldsymbol{x}, \lambda \cdot q). \tag{10}$$

Since scaling the model distribution $q$ by a positive constant $\lambda$ does not change the value of a homogeneous scoring rule, such homogeneity allows us to evaluate it without computing the intractable partition function of an energy-based distribution. Thus, strictly proper and homogeneous scoring rules are natural candidates for new training objectives of EBMs.

**Example 1.** *A notable example of scoring rules is the widely used logarithm score: $S(\boldsymbol{x}, q) = \log q(\boldsymbol{x})$. The associated generalized entropy is the negative Shannon entropy: $G(p) = \mathbb{E}_{p(\boldsymbol{x})}[\log p(\boldsymbol{x})]$, and the associated Bregman divergence is the KL divergence: $D(p, q) = \mathbb{E}_{p(\boldsymbol{x})}[\log(p(\boldsymbol{x})/q(\boldsymbol{x}))]$. From Definitions 1 and 2, we know that the logarithm score is strictly proper but not homogeneous. Specifically, for a $\boldsymbol{\theta}$-parametrized energy-based distribution $q_{\boldsymbol{\theta}} = \overline{q}_{\boldsymbol{\theta}}/Z_{\boldsymbol{\theta}}$, since $S(\boldsymbol{x}, \overline{q}_{\boldsymbol{\theta}}) = S(\boldsymbol{x}, Z_{\boldsymbol{\theta}} \cdot q_{\boldsymbol{\theta}}) = S(\boldsymbol{x}, q_{\boldsymbol{\theta}}) + \log Z_{\boldsymbol{\theta}} \neq S(\boldsymbol{x}, q_{\boldsymbol{\theta}})$ and $\log Z_{\boldsymbol{\theta}}$ cannot be ignored during the optimization of $\boldsymbol{\theta}$, we need to use tailored methods such as contrastive divergence [37] or doubly dual embedding [11] to tackle the intractable partition function.*

## 3 Training EBMs by Maximizing Homogeneous Scoring Rules

In this section, we derive a new principle for training EBMs from the perspective of optimizing strictly proper homogeneous scoring rules. All proofs for this section can be found in Appendix B.

### 3.1 Pseudo-Spherical Scoring Rule

In this section, we introduce the pseudo-spherical scoring rule, which is a representative family of strictly proper homogeneous scoring rules that have great potentials for training deep energy-based models and allow flexible and convenient specification of modeling preferences, yet have not been explored before in the context of energy-based generative modeling.

**Definition 3** (Pseudo-Spherical Scoring Rule [76, 27]). *For $\gamma > 0$, the pseudo-spherical scoring rule is defined as:*

$$S(\boldsymbol{x}, q) := \frac{q(\boldsymbol{x})^{\gamma}}{\left(\int_{\mathcal{X}} q(\boldsymbol{y})^{\gamma+1} \mathrm{d}\boldsymbol{y}\right)^{\frac{\gamma}{\gamma+1}}} = \frac{\overline{q}(\boldsymbol{x})^{\gamma}}{\left(\int_{\mathcal{X}} \overline{q}(\boldsymbol{y})^{\gamma+1} \mathrm{d}\boldsymbol{y}\right)^{\frac{\gamma}{\gamma+1}}} = \left(\frac{\overline{q}(\boldsymbol{x})}{\|\overline{q}\|_{\gamma+1}}\right)^{\gamma} \tag{11}$$

*where $\|\overline{q}\|_{\gamma+1} := \left(\int_{\mathcal{X}} \overline{q}(\boldsymbol{y})^{\gamma+1} \mathrm{d}\boldsymbol{y}\right)^{\frac{1}{\gamma+1}}$.*

*The expected pseudo-spherical score under a reference distribution $p$ is defined as:*

$$S_{ps}(p, q) := \mathbb{E}_{p(\boldsymbol{x})}[S(\boldsymbol{x}, q)] = \frac{\mathbb{E}_{p(\boldsymbol{x})}[\overline{q}(\boldsymbol{x})^{\gamma}]}{\left(\int_{\mathcal{X}} \overline{q}(\boldsymbol{y})^{\gamma+1} \mathrm{d}\boldsymbol{y}\right)^{\frac{\gamma}{\gamma+1}}} \tag{12}$$

**Example 2.** *The classic spherical scoring rule [19] is a special case in the pseudo-spherical family, which corresponds to $\gamma = 1$:*

$$S(\boldsymbol{x}, q) = \frac{\overline{q}(\boldsymbol{x})}{\left(\int_{\mathcal{X}} \overline{q}(\boldsymbol{y})^2 \mathrm{d}\boldsymbol{y}\right)^{\frac{1}{2}}} = \frac{\overline{q}(\boldsymbol{x})}{\|\overline{q}\|_2} \tag{13}$$

The family of pseudo-spherical scoring rules is appealing because it introduces a different and principled way for assessing a probability forecast. For example, the spherical scoring rule has an interesting geometric interpretation. Suppose the sample space $\mathcal{X}$ contains $n$ mutually exclusive and exhaustive outcomes. Then a probability forecast can be represented as a vector $\boldsymbol{q} = (q_1, \dots, q_n)$. Let vector $\boldsymbol{p} = (p_1, \dots, p_n)$ represent the oracle probability forecast. The expected spherical score can be written as:

$$S(p, q) = \mathbb{E}_{p(\boldsymbol{x})}[S(\boldsymbol{x}, q)] = \frac{\sum_i p_i q_i}{\sqrt{\sum_i q_i^2}} = \|\boldsymbol{p}\|_2 \frac{\langle \boldsymbol{p}, \boldsymbol{q} \rangle}{\|\boldsymbol{p}\|_2 \|\boldsymbol{q}\|_2} = \|\boldsymbol{p}\|_2 \cos(\angle(\boldsymbol{p}, \boldsymbol{q})) \tag{14}$$

where $\langle \boldsymbol{p}, \boldsymbol{q} \rangle$ and $\angle(\boldsymbol{p}, \boldsymbol{q})$ denote the inner product and the angle between vectors $\boldsymbol{p}$ and $\boldsymbol{q}$ respectively. In other words, when we want to evaluate the expected spherical score of a probability forecast $\boldsymbol{q}$ under real data distribution $\boldsymbol{p}$ using samples, the angle between $\boldsymbol{p}$ and $\boldsymbol{q}$ is the sufficient statistics. Since we know that both $\boldsymbol{p}$ and $\boldsymbol{q}$ belong to the probability simplex $\mathcal{P} = \{\boldsymbol{v} | \sum_{\boldsymbol{x} \in \mathcal{X}} \boldsymbol{v}(\boldsymbol{x}) = 1 \text{ and } \forall \boldsymbol{x} \in \mathcal{X}, \boldsymbol{v}(\boldsymbol{x}) \geq 0.\}$, the expected score will be minimized if and only if the angle is zero, which implies $\boldsymbol{p} = \boldsymbol{q}$. More importantly, since all we need to do is to minimize the angle of

the deviation, we are allowed to scale $\boldsymbol{q}$ by a constant. Specifically, when $\boldsymbol{q}$ is an energy-based distribution $\boldsymbol{q} = \left( \frac{\exp(-E_1)}{\sum_i \exp(-E_i)}, \ldots, \frac{\exp(-E_n)}{\sum_i \exp(-E_i)} \right)$, we can instead evaluate and minimize the angle between data distribution $\boldsymbol{p}$ and the unnormalized distribution $\overline{\boldsymbol{q}} = (\exp(-E_1), \ldots, \exp(-E_n))$, since $\angle(\boldsymbol{p}, \boldsymbol{q}) = \angle(\boldsymbol{p}, \overline{\boldsymbol{q}})$. More generally, we have the following theorem to justify the use of pseudo-spherical scoring rules for training energy-based models:

**Theorem 1** ([26, 70]). *Pseudo-spherical scoring rule is strictly proper and homogeneous.*

As the original definition of pseudo-spherical scoring rule (Equation (11)) takes the form of a fraction, for computational considerations, in this paper we instead focus on optimizing its *composite scoring rule* (Definition 2.1 in [44]):

**Definition 4** ($\gamma$-score [20]). *For the expected pseudo-spherical score $S_{ps}(p, q)$ defined in Equation (12) with $\gamma > 0$, the expected $\gamma$-score is defined as:*

$$S_\gamma(p, q) := \frac{1}{\gamma} \log(S_{ps}(p, q)) = \frac{1}{\gamma} \log \left( \mathbb{E}_{p(\boldsymbol{x})}[\overline{q}(\boldsymbol{x})^\gamma] \right) - \log(\|\overline{q}\|_{\gamma+1}) \tag{15}$$

Since $\frac{1}{\gamma} \log(u)$ is strictly increasing in $u$, $S_\gamma(p, q)$ is a strictly proper homogeneous composite score:

$$\arg\max_{q \in \mathcal{P}} S_\gamma(p, q) = \arg\max_{q \in \mathcal{P}} \frac{1}{\gamma} \log(S_{ps}(p, q)) = \arg\max_{q \in \mathcal{P}} S_{ps}(p, q) = p. \tag{16}$$

### 3.2 Pseudo-Spherical Contrastive Divergence

Suppose we parametrize the energy-based model distribution as $q_{\boldsymbol{\theta}} \propto \overline{q}_{\boldsymbol{\theta}} = \exp(-E_{\boldsymbol{\theta}})$ and we want to minimize the negative $\gamma$-score in Equation (15):

$$\min_{\boldsymbol{\theta}} \mathcal{L}_\gamma(\boldsymbol{\theta}; p) = \min_{\boldsymbol{\theta}} -\frac{1}{\gamma} \log \left( \mathbb{E}_{p(\boldsymbol{x})}[\overline{q}_{\boldsymbol{\theta}}(\boldsymbol{x})^\gamma] \right) + \log(\|\overline{q}_{\boldsymbol{\theta}}\|_{\gamma+1}) \tag{17}$$

In the following theorem, we derive the gradient of $\mathcal{L}_\gamma(\boldsymbol{\theta}; p)$ with respect to $\boldsymbol{\theta}$:

**Theorem 2.** *The gradient of $\mathcal{L}_\gamma(\boldsymbol{\theta}; p)$ with respect to $\boldsymbol{\theta}$ can be written as:*

$$\nabla_{\boldsymbol{\theta}} \mathcal{L}_\gamma(\boldsymbol{\theta}; p) = -\frac{1}{\gamma} \nabla_{\boldsymbol{\theta}} \log \left( \mathbb{E}_{p(\boldsymbol{x})}[\exp(-\gamma E_{\boldsymbol{\theta}}(\boldsymbol{x}))] \right) - \mathbb{E}_{r_{\boldsymbol{\theta}}(\boldsymbol{x})}[\nabla_{\boldsymbol{\theta}} E_{\boldsymbol{\theta}}(\boldsymbol{x})] \tag{18}$$

*where the auxiliary distribution $r_{\boldsymbol{\theta}}$ is also an energy-based distribution defined as:*

$$r_{\boldsymbol{\theta}}(\boldsymbol{x}) := \frac{\overline{q}_{\boldsymbol{\theta}}(\boldsymbol{x})^{\gamma+1}}{\int_{\mathcal{X}} \overline{q}_{\boldsymbol{\theta}}(\boldsymbol{x})^{\gamma+1} \mathrm{d}\boldsymbol{x}} = \frac{\exp(-(\gamma+1)E_{\boldsymbol{\theta}}(\boldsymbol{x}))}{\int_{\mathcal{X}} \exp(-(\gamma+1)E_{\boldsymbol{\theta}}(\boldsymbol{x})) \mathrm{d}\boldsymbol{x}}.$$

In App. B.1, we provide two different ways to prove the above theorem. The first one is more straightforward and directly differentiates through the term $\log(\|\overline{q}_{\boldsymbol{\theta}}\|_{\gamma+1})$. The second one leverages a variational representation of $\log(\|\overline{q}_{\boldsymbol{\theta}}\|_{\gamma+1})$, where the optimal variational distribution happens to take an analytical form of $r_{\boldsymbol{\theta}}^*(\boldsymbol{x}) \propto \overline{q}_{\boldsymbol{\theta}}(\boldsymbol{x})^{\gamma+1}$, thus avoiding the minimax optimization in other variational frameworks such as [89, 11, 12]. The main challenge in maximizing $\gamma$-score is that it is generally intractable to exactly compute the gradient of the second term in Equation (15).

During training, estimating the second term of Equation (18) requires us to obtain samples from the auxiliary distribution $r_{\boldsymbol{\theta}} \propto \exp(-(\gamma+1)E_{\boldsymbol{\theta}})$, while at test time, we want to sample from the model distribution $q_{\boldsymbol{\theta}} \propto \exp(-E_{\boldsymbol{\theta}})$ that approximates the data distribution. Due to the restrict regularity conditions on the convergence of Langevin dynamics, in practice, we found it challenging to use the iterative sampling process in Equation (2) with a fixed number of transition steps and step size to produce samples from $r_{\boldsymbol{\theta}}$ and $q_{\boldsymbol{\theta}}$ simultaneously, as the temperature $\gamma + 1$ in $r_{\boldsymbol{\theta}}$ simply amounts to a linear rescaling to the energy function during training. Thus for generality, as in contrastive divergence [37, 17, 66] and $f$-EBM [89], we make the minimal assumption that we only have a sampling procedure to produce samples from $q_{\boldsymbol{\theta}}$ for both learning and inference procedures.

In this case, we can leverage the analytical form of $r_{\boldsymbol{\theta}}$ and *self-normalized importance sampling* [68] (which has been used to derive gradient estimators in other contexts such as importance weighted autoencoder [7, 18]) to obtain a consistent estimation of Equation (18):

---

**Algorithm 1** Pseudo-Spherical Contrastive Divergence.

---

1: **Input:** Empirical data distribution $p_{\text{data}}$. Pseudo-spherical scoring rule hyperparameter $\gamma$.
2: Initialize energy function $E_{\boldsymbol{\theta}}$.
3: **repeat**
4:     Draw a minibatch of samples $\{\boldsymbol{x}_1^+, \ldots, \boldsymbol{x}_N^+\}$ from $p_{\text{data}}$.
5:     Draw a minibatch of samples $\{\boldsymbol{x}_1^-, \ldots, \boldsymbol{x}_N^-\}$ from $q_{\boldsymbol{\theta}} \propto \exp(-E_{\boldsymbol{\theta}})$ (*e.g.*, using Langevin dynamics with a sample replay buffer).
6:     Update the energy function by stochastic gradient descent:

$$\nabla_{\boldsymbol{\theta}} \widehat{\mathcal{L}_{\gamma}^N(\boldsymbol{\theta}; p)} = -\nabla_{\boldsymbol{\theta}} \frac{1}{\gamma} \log \left( \frac{1}{N} \sum_{i=1}^{N} \exp(-\gamma E_{\boldsymbol{\theta}}(\boldsymbol{x}_i^+)) \right) - \frac{\sum_{i=1}^{N} \exp(-\gamma E_{\boldsymbol{\theta}}(\boldsymbol{x}_i^-)) \nabla_{\boldsymbol{\theta}} E_{\boldsymbol{\theta}}(\boldsymbol{x}_i^-)}{\sum_{i=1}^{N} \exp(-\gamma E_{\boldsymbol{\theta}}(\boldsymbol{x}_i^-))}$$

7: **until** Convergence

---

**Theorem 3.** *Let $\boldsymbol{x}_1^+, \ldots, \boldsymbol{x}_N^+$ be i.i.d. samples from $p(\boldsymbol{x})$ and $\boldsymbol{x}_1^-, \ldots, \boldsymbol{x}_N^-$ be i.i.d. samples from $q_{\boldsymbol{\theta}}(\boldsymbol{x}) \propto \exp(-E_{\boldsymbol{\theta}}(\boldsymbol{x}))$. Define the gradient estimator as:*

$$\nabla_{\boldsymbol{\theta}} \widehat{\mathcal{L}_{\gamma}^N(\boldsymbol{\theta}; p)} := -\nabla_{\boldsymbol{\theta}} \frac{1}{\gamma} \log \left( \frac{1}{N} \sum_{i=1}^{N} \exp(-\gamma E_{\boldsymbol{\theta}}(\boldsymbol{x}_i^+)) \right) - \frac{\sum_{i=1}^{N} \omega_{\boldsymbol{\theta}}(\boldsymbol{x}_i^-) \nabla_{\boldsymbol{\theta}} E_{\boldsymbol{\theta}}(\boldsymbol{x}_i^-)}{\sum_{i=1}^{N} \omega_{\boldsymbol{\theta}}(\boldsymbol{x}_i^-)} \quad (19)$$

*where the self-normalized importance weight $\omega_{\boldsymbol{\theta}}(\boldsymbol{x}_i^-) := \bar{r}_{\boldsymbol{\theta}}(\boldsymbol{x}_i^-)/\bar{q}_{\boldsymbol{\theta}}(\boldsymbol{x}_i^-) = \exp(-\gamma E_{\boldsymbol{\theta}}(\boldsymbol{x}_i^-))$. Then the gradient estimator converges to the true gradient (Equation (18)) in probability, i.e., $\forall \epsilon > 0$:*

$$\lim_{N \to \infty} \mathbb{P} \left( \left\| \nabla_{\boldsymbol{\theta}} \widehat{\mathcal{L}_{\gamma}^N(\boldsymbol{\theta}; p)} - \nabla_{\boldsymbol{\theta}} \mathcal{L}_{\gamma}(\boldsymbol{\theta}; p) \right\| \geq \epsilon \right) = 0.$$

We summarize the pseudo-spherical contrastive divergence (PS-CD) training procedure in Algorithm 1. In Appendix A, we also provide a simple PyTorch implementation for stochastic gradient descent (SGD) with the gradient estimator in Equation (19).

### 3.3 Connections to Maximum Likelihood Estimation and Extension to $\gamma < 0$

From Equation (9) in Section 2.3, we know that $\gamma$-score induces the following Bregman divergence (the divergence function associated with proper composite scoring rule is analogously defined in Def. 2.1 in [44]):

$$D_{\gamma}(p, q_{\boldsymbol{\theta}}) = S_{\gamma}(p, p) - S_{\gamma}(p, q_{\boldsymbol{\theta}})$$

and maximizing $\gamma$-score is equivalent to minimizing $D_{\gamma}(p, q_{\boldsymbol{\theta}})$. In the following lemma, we show that when $\gamma \to 0$, $D_{\gamma}(p, q_{\boldsymbol{\theta}})$ will recover the KL divergence between $p$ and $q_{\boldsymbol{\theta}}$, and the gradient of PS-CD will recover the gradient of CD.

**Lemma 1.** *When $\gamma \to 0$, we have:*

$$\lim_{\gamma \to 0} D_{\gamma}(p, q_{\boldsymbol{\theta}}) = D_{\text{KL}}(p \| q_{\boldsymbol{\theta}}); \quad \lim_{\gamma \to 0} \nabla_{\boldsymbol{\theta}} \mathcal{L}_{\gamma}(\boldsymbol{\theta}; p) = \nabla_{\boldsymbol{\theta}} \mathcal{L}_{\text{MLE}}(\boldsymbol{\theta}; p).$$

Inspired by [86, 56] that generalize Rényi divergence beyond its definition to negative orders, we now consider the extension of $\gamma$-score with $\gamma < 0$ (although it may not be strictly proper for these $\gamma$ values). The following lemma shows that maximizing such scoring rule amounts to maximizing a lower bound of logarithm score (MLE) with an additional Rényi entropy regularization.

**Lemma 2.** *When $-1 \leq \gamma < 0$, we have:*

$$S_{\gamma}(p, q) \leq \mathbb{E}_{p(\boldsymbol{x})}[\log q(\boldsymbol{x})] + \frac{\gamma}{\gamma + 1} \mathcal{H}_{\gamma+1}(q)$$

*where $\mathcal{H}_{\gamma+1}(q)$ is the Rényi entropy of order $\gamma + 1$.*

## 4 Theoretical Analysis

In this section, to gain a deeper understanding of our PS-CD algorithm and how the proposed estimator behaves, we analyze its sample complexity and convergence property under certain conditions. All the proofs for this section can be found in Appendix C.

## 4.1 Sample Complexity

We start with analyzing the sample complexity of the consistent gradient estimator in Equation (19), that is how fast it approaches the true gradient value. We first make the following assumption:

**Assumption 1.** *The energy function is bounded by $K$ and the gradient is bounded by $L$:*

$$\forall \boldsymbol{x} \in \mathcal{X}, \, \boldsymbol{\theta} \in \Theta, \, |E_{\boldsymbol{\theta}}(\boldsymbol{x})| \leq K, \, \|\nabla_{\boldsymbol{\theta}} E_{\boldsymbol{\theta}}(\boldsymbol{x})\| \leq L.$$

This assumption is typically easy to satisfy because in practice we always use a bounded sample space (*e.g.* normalizing images to [0,1] or truncated Gaussian) to ensure stability. For example, in image modeling experiments, we use $L_2$ regularization on the outputs of the energy function (hence bounded energy values), as well as normalized inputs and spectral normalization [60] for the neural network that realizes the energy function (hence Lipschitz continuous with bounded gradient).

With vector Bernstein inequality [47, 32], we have the following theorem showing a sample complexity of $O\left(\frac{\log(1/\delta)}{\epsilon^2}\right)$ such that the estimation error is less than $\epsilon$ with probability at least $1 - \delta$:

**Theorem 4.** *For any constants $\epsilon > 0$ and $\delta \in (0, 1)$, when the number of samples $N$ satisfies:*

$$N \geq \frac{32L^2 e^{8\gamma K} \left(1 + 4\log(2/\delta)\right)}{\epsilon^2}$$

*we have:*

$$\mathbb{P}\left(\left\|\nabla_{\boldsymbol{\theta}}\widehat{\mathcal{L}_{\gamma}^N(\boldsymbol{\theta}; p)} - \nabla_{\boldsymbol{\theta}}\mathcal{L}_{\gamma}(\boldsymbol{\theta}; p)\right\| \leq \epsilon\right) \geq 1 - \delta.$$

## 4.2 Convergence

Typically, convergence of SGD are analyzed for unbiased gradient estimators, while the gradient estimator in PS-CD is asymptotically consistent but biased. Building on the sample complexity bound above and prior theoretical works for analyzing SGD [8, 24], we analyze the convergence of PS-CD. For notational convenience, we use $\mathcal{L}(\boldsymbol{\theta})$ to denote the loss function $\mathcal{L}_{\gamma}(\boldsymbol{\theta}; p) = -S_{\gamma}(p, q_{\boldsymbol{\theta}})$. Besides Assumption 1, we further make the following assumption on the smoothness of $\mathcal{L}(\boldsymbol{\theta})$:

**Assumption 2.** *The loss function $\mathcal{L}(\boldsymbol{\theta})$ is $M$-smooth (with $M > 0$):*

$$\forall \boldsymbol{\theta}_1, \boldsymbol{\theta}_2 \in \Theta, \, \|\nabla\mathcal{L}(\boldsymbol{\theta}_1) - \nabla\mathcal{L}(\boldsymbol{\theta}_2)\| \leq M\|\boldsymbol{\theta}_1 - \boldsymbol{\theta}_2\|.$$

This is a common assumption for analyzing first-order optimization methods, which is also used in [24, 8]. Also this is a relatively mild assumption since we do not require the loss function to be convex in $\boldsymbol{\theta}$. Since in non-convex optimization, the convergence criterion is typically measured by gradient norm, following [64, 24], we use $\|\nabla\mathcal{L}(\boldsymbol{\theta})\| \leq \xi$ to judge whether a solution $\boldsymbol{\theta}$ is approximately a stationary point.

**Theorem 5.** *For any constants $\alpha \in (0, 1)$ and $\delta \in (0, 1)$, suppose that the step sizes satisfy $\eta_t < 2(1 - \alpha)/M$ and the sample size $N_t$ used for estimating $\widehat{\boldsymbol{g}}_t$ is sufficiently large (satisfying Equation (36)). Let $\mathcal{L}^*$ denote the minimum value of $\mathcal{L}(\boldsymbol{\theta})$. Then with probability at least $1 - \delta$, the output of Algorithm 2 (in Appendix C.2), $\widehat{\boldsymbol{\theta}}$, satisfies (constant $C := \alpha M L^2 e^{4\gamma K}$):*

$$\mathbb{E}[\|\nabla\mathcal{L}(\widehat{\boldsymbol{\theta}})\|^2] < \frac{2(\mathcal{L}(\boldsymbol{\theta}_1) - \mathcal{L}^*) + 12C\sum_{t=1}^{T}\eta_t^2}{\sum_{t=1}^{T}(2(1 - \alpha)\eta_t - M\eta_t^2)}$$

The above theorem implies the following corollary, which shows a typical convergence rate of $O(1/\sqrt{T})$ for non-convex optimization problems:

**Corollary 1.** *Under the conditions in Theorem 5 except that we use constant step sizes: $\eta_t = \min\{(1 - \alpha)/M, 1/\sqrt{T}\}$ for $t = 1, \ldots, T$. Then with probability at least $1 - \delta$, we have (constant $C := \alpha M L^2 e^{4\gamma K}$):*

$$\mathbb{E}[\|\nabla\mathcal{L}(\widehat{\boldsymbol{\theta}})\|^2] < \frac{2M(\mathcal{L}(\boldsymbol{\theta}_1) - \mathcal{L}^*)}{(1 - \alpha)^2 T} + \frac{2(\mathcal{L}(\boldsymbol{\theta}_1) - \mathcal{L}^*) + 12C}{(1 - \alpha)\sqrt{T}}$$

In Appendix C.2, we discuss more on the strongly convex (Theorem 7) and convex cases (Theorem 8).

# 5 Related Work

**Direct KL Minimization.** Under the "analysis by synthesis" scheme, [37] proposed Contrastive Divergence (CD), which estimates the gradient of the log-partition function (arising from KL) using samples from some MCMC procedure. To improve the mixing time of MCMC, [17] proposed to employ Persistent CD and a replay buffer to store intermediate samples from Markov chains throughout training, and [66] proposed to learn non-convergent short-run MCMC. Both approaches (long-run and short-run MCMC) work well with PS-CD in our experiments. PS-CD may also benefit from recent advances on unbiased MCMC [40, 73], which we leave as interesting future work.

**Fenchel Duality.** By exploiting the primal-dual view of KL, recent works [11, 12, 2] proposed to cast maximum likelihood training of EBMs as minimax problems, which introduce a dual sampler and are approximately solved by alternating gradient descent ascent updates. Along this line, to allow flexible modeling preferences, [89] proposed $f$-EBM to enable the use of any $f$-divergence to train EBMs, which also relies on minimax optimization. By contrast, in this work, we leverage the analytical form of the optimal variational distribution and self-normalized importance sampling to reach a framework that requires no adversarial training and has no additional computational cost compared to CD while allowing flexible modeling preferences. Besides convenient optimization, PS-CD and $f$-EBM trains EBMs with two different families of divergences (hence complementary) with KL being the only shared one, since any pseudo-spherical scoring rule corresponds to a Bregman divergence (Section 2.3) and the only member in $f$-divergence that is also Bregman divergence is $\alpha$-divergence (with KL as special case) (Theorem 4 in [1]).

**Homogeneous Scoring Rules.** [84] proposed to learn unnormalized statistical models on discrete sample space by maximizing $\gamma$-score, which uses empirical data distribution ($\widehat{p}(\boldsymbol{x}) = n_{\boldsymbol{x}}/n$, where $n_{\boldsymbol{x}}$ is the number of appearance of $\boldsymbol{x}$ in the dataset and $n$ is the total number of data) as a surrogate to the real data density $p$ and relies on a localization trick to bypass the computation of $\|q_{\boldsymbol{\theta}}\|_{\gamma+1}$. Consequently, it is only amenable to finite discrete sample space such as natural language [51], whereas PS-CD is applicable to any unnormalized probabilistic model in continuous domains. Another popular homogeneous scoring rule is the Hyvärinen score, which gives rise to the score matching objective [39] for EBM training. However, score matching and its variants [87, 82] have difficulties in low data density regions and do not perform well in practice when training EBMs on high-dimensional datasets [80]. Moreover, since the score matching objective involves the Hessian of log-density functions that is generally expensive to compute [58], methods such as approximate propagation [45], curvature propagation [58] and sliced score matching [83] are needed to approximately compute the trace of the Hessian.

**Noise Contrastive Estimation.** Another principle for learning EBMs is Noise Contrastive Estimation (NCE) [34], where an EBM is learned by contrasting a prescribed noise distribution with tractable density against the unknown data distribution. Using various Bregman divergences, NCE can be generalized to a family of different loss functionals [33, 85]. However, finding an appropriate noise distribution for NCE is highly non-trivial. In practice, NCE typically works well in conjunction with a carefully-designed noise distribution such as context-dependent noise distribution [41] or joint learning of a flow-based noise distribution [21].

In this work, we focus on generalizing maximum likelihood by deriving novel training objectives for EBMs without involving auxiliary models (*e.g.*, the variational function in [89], the flow-based noise distribution in [21] and the amortized sampler in [50, 12, 29]).

# 6 Experiments

In this section, we demonstrate the effectiveness of PS-CD on several 1-D and 2-D synthetic datasets as well as commonly used image datasets.

**Setup.** The 2-D synthetic datasets include Cosine, Swiss Roll, Moon, Mixture of Gaussian, Funnel and Rings, which cover different modalities and geometries (see Figure 2 in App. D.1 for illustration). To test the practical usefulness, we use MNIST [54], CIFAR-10 [48] and CelebA [57] in our experiments for modeling natural images. Following [80], for CelebA, we first center-crop the images to $140 \times 140$, then resize them to $64 \times 64$. More experimental details about the data processing, model architectures, sampling strategies and additional experimental results can be found in App. D.

**Effects of Different $\gamma$ Values.** To illustrate the modeling flexibility brought by PS-CD and provide insights on the effects of different $\gamma$ values, we first conduct a 1-D synthetic experiment similar to the one in [89]. As shown in Figure 1, when fitting a mixture of Gaussian with a single Gaussian (*i.e.*, model mis-specification case), the family of PS-CD offers flexible tradeoffs between quality and diversity (*i.e.*, mode collapse vs. mode coverage). Although in the well-specified case these objectives induce the same optimal solution, in this example, we can see that a larger $\gamma$ leads to higher entropy. More importantly, compared to $f$-EBM [89]

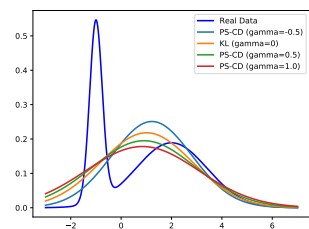

Figure 1: The effects of different $\gamma$ values when fitting a mixture of Gaussian with a single Gaussian.

that also provides similar modeling flexibility and includes CD as a special case, PS-CD does not require expensive and unstable minimax optimization (no additional computational cost compared to CD). In App. D.2, we further visualize the objective landscapes for different $\gamma$ values. As shown in Figure 3 and 4, when the model is well-specified, different objectives will induce the same optimal solution since they are strictly proper; when the model is mis-specified (corresponding to practical scenarios), different objectives will exhibit different modeling preferences.

Furthermore, to better demonstrate the modeling flexibility brought by PS-CD in high-dimensional case, we conduct experiments on CIFAR-10 using a set of indicative and reliable metrics (Density, Coverage, Precision, Recall) proposed by [61] to evaluate the effects of $\gamma$ from various perspectives. Please refer to App. D.3 for experimental results (Table 4) and detailed discussions.

**2-D Synthetic Data.** For quantitative evaluation of the 2-D synthetic data experiments, we follow [79] and report the maximum mean discrepancy (MMD, [5]) between the generated samples and validation samples in Table 3 in App. D.1, which demonstrates that PS-CD outperforms its CD counterpart on all but the Funnel dataset. From the histograms of samples shown in Figure 2 in App. D.1, we can also have similar observations. For example, CD fails to place high densities in the center of the right mode in MoG, while PS-CD places the modes correctly.

**Image Generation.** In Figure 5 in App.D.4, we show MNIST, CIFAR-10 and CelebA samples produced by PS-CD (with $\gamma = 1.0$), which demonstrate that our approach can produce highly realistic images with simple model architectures. As suggested in [17], we use Fréchet Inception Distance (FID) [36] as the quantitative evaluation metric for CIFAR-10 and CelebA, as Langevin dynamics converge to local minima that artificially inflate Inception Score [77]. From Table 1[1], we can see that various members (different $\gamma$ values) in the family of PS-CD can outperform CD significantly, and more surprisingly, PS-CD also shows competitive performance to the recently proposed $f$-EBMs, without requiring expensive minimax optimization. While our method currently does not outperform the state-of-the-art image generation methods such as improved denoising score matching [81], which relies on carefully selected noise schedule and specially designed

Table 1: FID scores for CIFAR-10 conditional and CelebA unconditional image generation. We list comparisons with results reported by CD [17], Noise-Conditional Score Network (NCSN) [81] and $f$-EBMs [89]. $\gamma = 1.0$ corresponds to maximizing spherical scoring rule (Example 2).

| Method | FID |
|---|---|
| **CIFAR-10 ($32 \times 32$) Conditional** | |
| Contrastive Divergence (KL) | 37.90 |
| $f$-EBM (KL) | 37.36 |
| $f$-EBM (Reverse KL) | 33.25 |
| $f$-EBM (Squared Hellinger) | 32.19 |
| $f$-EBM (Jensen Shannon) | 30.86 |
| Pseudo-Spherical CD ($\gamma = 2.0$) | 33.19 |
| Pseudo-Spherical CD ($\gamma = 1.0$) | **29.78** |
| Pseudo-Spherical CD ($\gamma = 0.5$) | 35.02 |
| Pseudo-Spherical CD ($\gamma = -0.5$) | **27.95** |
| **CelebA ($64 \times 64$)** | |
| Contrastive Divergence (KL) | 26.10 |
| NCSN (w/o denoising) | 26.89 |
| NCSN (w/ denoising) | 25.30 |
| NCSNv2 (w/o denoising) | 28.86 |
| NCSNv2 (w/ denoising) | **10.23** |
| Pseudo-Spherical CD ($\gamma = 1.0$) | **24.76** |
| Pseudo-Spherical CD ($\gamma = -0.5$) | **20.35** |

noise-dependent score network (modified U-Net architecture, hence not directly comparable to our results), we think that our work opens up a new research direction by bridging statistical decision theory (homogeneous proper scoring rules) and deep energy-based generative modeling. Moreover,

---

[1]For CelebA dataset, we reproduced the short-run MCMC method [66] using our code base. Moreover, the $f$-EBM paper only reported results on CelebA 32x32 and we empirically found it is not comparable to our method in CelebA 64x64 (higher resolution), indicating better scalability of PS-CD to high-dimensional case.

Table 2: Robustness to data contamination on Gaussian datasets. The data distribution is $\mathcal{N}(-1, 0.5)$ and the contamination distribution is $\mathcal{N}(2, 0.05)$. We measure the KL divergence between clean target distribution $p$ and converged model distribution $q_\theta$, $D_{\mathrm{KL}}(p\|q_\theta)$.

| Contamination Ratio | CD | PS-CD ($\gamma = 0.5$) | PS-CD ($\gamma = 1.0$) | PS-CD ($\gamma = 2.0$) |
|---|---|---|---|---|
| 0.01 | 0.0067 | 1e-5 | 1e-7 | 1e-6 |
| 0.05 | 0.0851 | 0.00027 | 1.6e-6 | 0.00011 |
| 0.1 | 0.1979 | 0.00173 | 1.86e-6 | 0.00012 |
| 0.2 | 0.3869 | 0.1858 | 6.4e-6 | 0.00017 |
| 0.3 | 0.5438 | 0.5429 | 0.3118 | 0.00029 |

under the setting of simple model architectures and the same hyperparameter configuration (*e.g.*, batch size, learning rate, network structure, etc.), our empirical results suggest clear superiority of PS-CD over traditional CD and recent $f$-EBMs.

**OOD Detection & Robustness to Data Contamination.** We further test our methods on out-of-distribution (OOD) detection tasks. For the conditional CIFAR-10 model, we follow the evaluation protocol in [17] and use $s(\boldsymbol{x}) = \max_{y\in\mathcal{Y}} -E(\boldsymbol{x}, y)$ as the score for detecting outliers. We use SVHN [65], Textures [9], Uniform/Gaussian Noise, CIFAR-10 Linear Interpolation and CelebA as the OOD datasets. We summarize the results in Table 5 in App. D.6, from which we can see that PS-CD consistently outperforms CD and other likelihood-based models.

Inspired by the OOD detection performance and previous work on robust parameter estimation under data contamination [43], we further test the robustness of CD and PS-CD on both synthetic and natural image datasets. Specifically, suppose $p(\boldsymbol{x})$ is the underlying data distribution and there is another contamination distribution $\omega(\boldsymbol{x})$, e.g. uniform noise. In generative modeling under data contamination, our model observes i.i.d. samples from the contaminated distribution $\tilde{p}(\boldsymbol{x}) = cp(\boldsymbol{x}) + (1 - c)\omega(\boldsymbol{x})$, where $1 - c \in [0, 1/2)$ is the contamination ratio. A theoretical advantage of pseudo-spherical score is its robustness to data contamination: the optimal solution of $\min_\theta S_{\mathrm{ps}}(\tilde{p}, q_\theta)$ is close to that of $\min_\theta S_{\mathrm{ps}}(p, q_\theta)$ under some conditions (e.g. the density of $\omega(\boldsymbol{x})$ mostly lies in the region for which the target density $p(\boldsymbol{x})$ is small) [20, 43]. From Table 2, we can see that CD suffers from data contamination severely: as the contamination ratio increases, the performance degrades drastically. By contrast, PS-CD shows good robustness against data contamination and a larger $\gamma$ leads to better robustness. For example, PS-CD with $\gamma = 1.0$ can properly approximate the target distribution when the contamination ratio is 0.2, while PS-CD with $\gamma = 2.0$ can do so when the contamination ratio is up to 0.3.

We conduct similar experiments on MNIST and CIFAR-10 datasets, where we use uniform noise as the contamination distribution and the contamination ratio is 0.1 (i.e. 10% images in the training set are replaced with random noise). After a warm-up pretraining[2] (when the model has some OOD detection ability), we train the model with the contaminated data and measure the training progress. We observe that CD gradually generates more random noise and diverge after a few training steps, while PS-CD is very robust. As shown in Table 6 in App. D.6, for a slightly pre-trained unconditional CIFAR-10 model (a simple 5-layer CNN with FID of 68.77), we observe that the performance of CD degrades drastically in terms of FID, while PS-CD can continuously improve the model even using the contaminated data. We provide visualizations and theoretical explanations in App. D.6.

## 7   Conclusion

From the perspective of maximizing strictly proper homogeneous scoring rules, we propose pseudo-spherical contrastive divergence (PS-CD) to generalize maximum likelihood estimation of energy-based models. Different from prior works that involve joint training of auxiliary models [89, 21, 50, 12, 29], PS-CD allows us to specify flexible modeling preferences without additional computational cost compared to contrastive divergence. We provide a theoretical analysis on the sample complexity and convergence property of the proposed method, as well as its connection to maximum likelihood. Finally, we demonstrate the effectiveness of PS-CD with extensive experiments on both synthetic data and commonly used image datasets.

---

[2]Note that it is impossible for a randomly initialized model to be robust to data contamination since without additional inductive bias, it will simply treat the contaminated distribution $\tilde{p}(\boldsymbol{x})$ as the target.

## Acknowledgements

This research was supported by NSF(#1651565, #1522054, #1733686), ONR (N000141912145), AFOSR (FA95501910024), ARO (W911NF-21-1-0125) and Sloan Fellowship.

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
