# A Implementation of PS-CD Algorithm

In the following, we provide a simple PyTorch [71] implementation of Step 6 in Algorithm 1, where we directly differentiate through $\log(\text{mean}(\exp(\cdot)))$ for the first term and perform stop gradient operation on the self-normalized importance weight $(\exp(-\gamma E_{\boldsymbol{\theta}}(\boldsymbol{x}_i^-)))$ for the second term. Mathematically, these two implementations are equivalent and we use different implementations here to illustrate the difference between their derivations. We further apply $L_2$ regularization on the outputs of the energy function to stabilize training.

```python
def logmeanexp(inputs):
    // Stable version log(mean(exp(inputs)))
    return inputs.max() + (inputs - inputs.max()).exp().mean().log()

def softmax(inputs):
    // Stable version softmax(inputs)
    exp_inputs = torch.exp(inputs - inputs.max())
    return exp_inputs / exp_inputs.sum()

def update_step(x_pos, x_neg, model_e, optim_e, l2_reg, gamma):
    // x_pos and x_neg are samples from p_data and q_theta.
    // model_e is the neural network for the energy function.
    // optim_e is the optimizer for model_e
    // e.g. optim_e = torch.optim.Adam(model_e.parameters())
    e_pos, e_neg = model_e(x_pos), model_e(x_neg)
    importance_weight = softmax(- gamma * e_neg)
    loss = - 1 / gamma * logmeanexp(- gamma * e_pos) - \
            torch.sum(e_neg * importance_weight.detach())
    loss += l2_reg * ((e_pos ** 2).mean() + (e_neg ** 2).mean())
    optim_e.zero_grad()
    loss_e.backward()
    optim_e.step()
```

# B Proofs for Section 3

## B.1 Proof for Theorem 2

In this section, we provide two different ways to prove Theorem 2. The first one is more straightforward and directly differentiates through the term $\log(\|\bar{q}_{\boldsymbol{\theta}}\|_{\gamma+1})$. The second one leverages a variational representation of $\log(\|\bar{q}_{\boldsymbol{\theta}}\|_{\gamma+1})$, where the optimal variational distribution happens to take an analytical form of $r_{\boldsymbol{\theta}}^*(\boldsymbol{x}) \propto \bar{q}_{\boldsymbol{\theta}}(\boldsymbol{x})^{\gamma+1}$, thus avoiding the minimax optimization in other variational frameworks for KL and $f$-divergences [89, 11, 12] and revealing the elegance of PS-CD.

**Theorem 2.** *For an energy-based distribution $q_{\boldsymbol{\theta}} \propto \bar{q}_{\boldsymbol{\theta}} = \exp(-E_{\boldsymbol{\theta}})$, the gradient of the negative $\gamma$-score $L_{\gamma}(\boldsymbol{\theta}; p) = -S_{\gamma}(p, q_{\boldsymbol{\theta}})$ with respect to $\boldsymbol{\theta}$ can be written as:*

$$\nabla_{\boldsymbol{\theta}} \mathcal{L}_{\gamma}(\boldsymbol{\theta}; p) = -\frac{1}{\gamma} \nabla_{\boldsymbol{\theta}} \log \left( \mathbb{E}_{p(\boldsymbol{x})}[\exp(-\gamma E_{\boldsymbol{\theta}}(\boldsymbol{x}))] \right) - \mathbb{E}_{r_{\boldsymbol{\theta}}(\boldsymbol{x})}[\nabla_{\boldsymbol{\theta}} E_{\boldsymbol{\theta}}(\boldsymbol{x})] \tag{20}$$

*where the auxiliary distribution $r_{\boldsymbol{\theta}}$ is also an energy-based distribution defined as:*

$$r_{\boldsymbol{\theta}}(\boldsymbol{x}) := \frac{\bar{q}_{\boldsymbol{\theta}}(\boldsymbol{x})^{\gamma+1}}{\int_{\mathcal{X}} \bar{q}_{\boldsymbol{\theta}}(\boldsymbol{x})^{\gamma+1} \mathrm{d}\boldsymbol{x}} = \frac{\exp(-(\gamma+1)E_{\boldsymbol{\theta}}(\boldsymbol{x}))}{\int_{\mathcal{X}} \exp(-(\gamma+1)E_{\boldsymbol{\theta}}(\boldsymbol{x})) \mathrm{d}\boldsymbol{x}}.$$

*Proof.* **First proof: direct differentiation.** From Equation (17), we have:

$$\nabla_{\boldsymbol{\theta}} \mathcal{L}_{\gamma}(\boldsymbol{\theta}; p) = \nabla_{\boldsymbol{\theta}} \left( -\frac{1}{\gamma} \log \left( \mathbb{E}_{p(\boldsymbol{x})}[\bar{q}_{\boldsymbol{\theta}}(\boldsymbol{x})^{\gamma}] \right) + \log(\|\bar{q}_{\boldsymbol{\theta}}\|_{\gamma+1}) \right)$$

$$= -\frac{1}{\gamma} \nabla_{\boldsymbol{\theta}} \log \left( \mathbb{E}_{p(\boldsymbol{x})}[\exp(-\gamma E_{\boldsymbol{\theta}}(\boldsymbol{x}))] \right) + \frac{1}{\gamma+1} \nabla_{\boldsymbol{\theta}} \log \left( \int_{\mathcal{X}} \exp(-(\gamma+1)E_{\boldsymbol{\theta}}(\boldsymbol{x})) \mathrm{d}\boldsymbol{x} \right)$$

$$= -\frac{1}{\gamma} \nabla_{\boldsymbol{\theta}} \log \left( \mathbb{E}_{p(\boldsymbol{x})}[\exp(-\gamma E_{\boldsymbol{\theta}}(\boldsymbol{x}))] \right) + \frac{1}{\gamma+1} \frac{\int_{\mathcal{X}} \exp(-(\gamma+1)E_{\boldsymbol{\theta}}(\boldsymbol{x})) \cdot (-(\gamma+1)\nabla_{\boldsymbol{\theta}} E_{\boldsymbol{\theta}}(\boldsymbol{x})) \mathrm{d}\boldsymbol{x}}{\int_{\mathcal{X}} \exp(-(\gamma+1)E_{\boldsymbol{\theta}}(\boldsymbol{x})) \mathrm{d}\boldsymbol{x}}$$

$$= -\frac{1}{\gamma} \nabla_{\boldsymbol{\theta}} \log \left( \mathbb{E}_{p(\boldsymbol{x})}[\exp(-\gamma E_{\boldsymbol{\theta}}(\boldsymbol{x}))] \right) - \int_{\mathcal{X}} \frac{\exp(-(\gamma+1)E_{\boldsymbol{\theta}}(\boldsymbol{x}))}{\int_{\mathcal{X}} \exp(-(\gamma+1)E_{\boldsymbol{\theta}}(\boldsymbol{y})) \mathrm{d}\boldsymbol{y}} \nabla_{\boldsymbol{\theta}} E_{\boldsymbol{\theta}}(\boldsymbol{x}) \mathrm{d}\boldsymbol{x}$$

$$= -\frac{1}{\gamma} \nabla_{\boldsymbol{\theta}} \log \left( \mathbb{E}_{p(\boldsymbol{x})}[\exp(-\gamma E_{\boldsymbol{\theta}}(\boldsymbol{x}))] \right) - \mathbb{E}_{r_{\boldsymbol{\theta}}(\boldsymbol{x})}[\nabla_{\boldsymbol{\theta}} E_{\boldsymbol{\theta}}(\boldsymbol{x})]$$

**Second proof: a variational representation with optimal variational distribution taking analytical form.** The main challenge is that the $\log \|\bar{q}_{\boldsymbol{\theta}}\|_{\gamma+1}$ term in $\mathcal{L}_{\gamma}(\boldsymbol{\theta}; p)$ is generally intractable to compute. To solve this issue, we introduce the following variational representation:

**Lemma 1.** *Let $\Delta_{\mathcal{X}}$ denote the set of all normalized probability density functions on sample space $\mathcal{X}$. With Fenchel duality, we have:*

$$(\gamma+1)\log(\|q\|_{\gamma+1}) = \log \left( \int_{\mathcal{X}} \bar{q}(\boldsymbol{x})^{\gamma+1} \mathrm{d}\boldsymbol{x} \right) = \max_{r \in \Delta_{\mathcal{X}}} \int_{\mathcal{X}} r(\boldsymbol{x}) \log \left( \bar{q}(\boldsymbol{x})^{\gamma+1} \right) \mathrm{d}\boldsymbol{x} - \int_{\mathcal{X}} r(\boldsymbol{x}) \log r(\boldsymbol{x}) \mathrm{d}\boldsymbol{x} \tag{21}$$

*where the maximum is attained at $r^*(\boldsymbol{x}) = \frac{\bar{q}(\boldsymbol{x})^{\gamma+1}}{\int_{\mathcal{X}} \bar{q}(\boldsymbol{x})^{\gamma+1} \mathrm{d}\boldsymbol{x}}$.*

*Proof.* With Jensen's inequality, we have:

$$\log \left( \int_{\mathcal{X}} \bar{q}(\boldsymbol{x})^{\gamma+1} \mathrm{d}\boldsymbol{x} \right) = \log \left( \int_{\mathcal{X}} r(\boldsymbol{x}) \frac{\bar{q}(\boldsymbol{x})^{\gamma+1}}{r(\boldsymbol{x})} \mathrm{d}\boldsymbol{x} \right) \geq \int_{\mathcal{X}} r(\boldsymbol{x}) \log \left( \frac{\bar{q}(\boldsymbol{x})^{\gamma+1}}{r(\boldsymbol{x})} \right) \mathrm{d}\boldsymbol{x}$$

The equality holds if and only if:

$$r^*(\boldsymbol{x}) \propto \bar{q}(\boldsymbol{x})^{\gamma+1}$$

As $r^* \in \Delta_{\mathcal{X}}$ is a normalized distribution, we have:

$$r^*(\boldsymbol{x}) = \frac{\bar{q}(\boldsymbol{x})^{\gamma+1}}{\int_{\mathcal{X}} \bar{q}(\boldsymbol{x})^{\gamma+1} \mathrm{d}\boldsymbol{x}}$$

$\square$

Now, suppose we parametrize the energy-based model distribution as $\overline{q}_{\boldsymbol{\theta}} = \exp(-E_{\boldsymbol{\theta}})$ and the variational distribution as $r_{\boldsymbol{\psi}}$. By plugging the variational representation in Lemma 1 into $\gamma$-score (Equation (15)), we obtain the following minimax formulation to minimize the negative $\gamma$-score:

$$\boldsymbol{\psi}^*(\boldsymbol{\theta}) = \arg\max_{\boldsymbol{\psi}} \mathcal{L}_{\gamma}(\boldsymbol{\theta}, \boldsymbol{\psi}; p) \quad \boldsymbol{\theta}^* = \arg\min_{\boldsymbol{\theta}} \max_{\boldsymbol{\psi}} \mathcal{L}_{\gamma}(\boldsymbol{\theta}, \boldsymbol{\psi}; p) = \arg\min_{\boldsymbol{\theta}} \mathcal{L}_{\gamma}(\boldsymbol{\theta}, \boldsymbol{\psi}^*(\boldsymbol{\theta}); p)$$

where the game value function $L_{\gamma}(\boldsymbol{\theta}, \boldsymbol{\psi}; p)$ is defined as (s.t. $L_{\gamma}(\boldsymbol{\theta}, \boldsymbol{\psi}^*(\boldsymbol{\theta}); p) = -S_{\gamma}(p, q_{\boldsymbol{\theta}})$):

$$L_{\gamma}(\boldsymbol{\theta}, \boldsymbol{\psi}; p) = \underbrace{-\frac{1}{\gamma} \log\left(\mathbb{E}_{p(\boldsymbol{x})}[\overline{q}_{\boldsymbol{\theta}}(\boldsymbol{x})^{\gamma}]\right)}_{L_1(\boldsymbol{\theta})} + \underbrace{\frac{1}{\gamma+1}\left(\mathbb{E}_{r_{\boldsymbol{\psi}}(\boldsymbol{x})}[\log\left(\overline{q}_{\boldsymbol{\theta}}(\boldsymbol{x})^{\gamma+1}\right)] - \mathbb{E}_{r_{\boldsymbol{\psi}}(\boldsymbol{x})}[\log r_{\boldsymbol{\psi}}(x)]\right)}_{L_2(\boldsymbol{\theta}, \boldsymbol{\psi})}$$

By Lemma 1, we know that $r_{\boldsymbol{\psi}^*(\boldsymbol{\theta})} \propto \overline{q}_{\boldsymbol{\theta}}^{\gamma+1}$.

The first term in Equation (20) is simply $\nabla_{\boldsymbol{\theta}} L_1(\boldsymbol{\theta})$. For the second term, since $L_2(\boldsymbol{\theta}, \boldsymbol{\psi})$ is a function of both $\boldsymbol{\theta}$ and $\boldsymbol{\psi}$, and the optimal variational parameter $\boldsymbol{\psi}^*(\boldsymbol{\theta})$ depends on $\boldsymbol{\theta}$, the total derivative of $L_2(\boldsymbol{\theta}, \boldsymbol{\psi}^*(\boldsymbol{\theta}))$ with respect to $\boldsymbol{\theta}$ is:

$$\frac{\mathrm{d}L_2(\boldsymbol{\theta}, \boldsymbol{\psi}^*(\boldsymbol{\theta}))}{\mathrm{d}\boldsymbol{\theta}} = \frac{\partial L_2(\boldsymbol{\theta}, \boldsymbol{\psi}^*(\boldsymbol{\theta}))}{\partial \boldsymbol{\theta}} + \frac{\partial L_2(\boldsymbol{\theta}, \boldsymbol{\psi}^*(\boldsymbol{\theta}))}{\partial \boldsymbol{\psi}^*(\boldsymbol{\theta})} \frac{\mathrm{d}\boldsymbol{\psi}^*(\boldsymbol{\theta})}{\mathrm{d}\boldsymbol{\theta}}$$

Because $\boldsymbol{\psi}^*(\boldsymbol{\theta})$ is the optimum of $L_2(\boldsymbol{\theta}, \boldsymbol{\psi})$ (Lemma 1), the second term in above equation is zero:

$$\frac{\partial L_2(\boldsymbol{\theta}, \boldsymbol{\psi}^*(\boldsymbol{\theta}))}{\partial \boldsymbol{\psi}^*(\boldsymbol{\theta})} = \frac{1}{\gamma+1}\left(\int_{\mathcal{X}} \nabla_{\boldsymbol{\psi}} r_{\boldsymbol{\psi}}(\boldsymbol{x}) \log\left(\overline{q}_{\boldsymbol{\theta}}(\boldsymbol{x})^{\gamma+1}\right) - \nabla_{\boldsymbol{\psi}} r_{\boldsymbol{\psi}}(\boldsymbol{x}) \log r_{\boldsymbol{\psi}}(\boldsymbol{x}) - \frac{r_{\boldsymbol{\psi}}(\boldsymbol{x})}{r_{\boldsymbol{\psi}}(\boldsymbol{x})} \nabla_{\boldsymbol{\psi}} r_{\boldsymbol{\psi}}(\boldsymbol{x}) \mathrm{d}\boldsymbol{x}\right)\Bigg|_{\boldsymbol{\psi}=\boldsymbol{\psi}^*(\boldsymbol{\theta})}$$

$$= \frac{1}{\gamma+1}\left(\int_{\mathcal{X}} (\log Z_{\boldsymbol{\psi}^*(\boldsymbol{\theta})} - 1)\nabla_{\boldsymbol{\psi}} r_{\boldsymbol{\psi}}(\boldsymbol{x})\mathrm{d}\boldsymbol{x}\right)\Bigg|_{\boldsymbol{\psi}=\boldsymbol{\psi}^*(\boldsymbol{\theta})}$$

$$= \frac{1}{\gamma+1}\left((\log Z_{\boldsymbol{\psi}^*(\boldsymbol{\theta})} - 1)\nabla_{\boldsymbol{\psi}} \int_{\mathcal{X}} r_{\boldsymbol{\psi}}(\boldsymbol{x})\mathrm{d}\boldsymbol{x}\right) = 0$$

where $Z_{\boldsymbol{\psi}^*(\boldsymbol{\theta})}$ is the partition function of $r_{\boldsymbol{\psi}^*(\boldsymbol{\theta})}$.

Thus we have:

$$\frac{\mathrm{d}L_2(\boldsymbol{\theta}, \boldsymbol{\psi}^*(\boldsymbol{\theta}))}{\mathrm{d}\boldsymbol{\theta}} = \frac{\partial L_2(\boldsymbol{\theta}, \boldsymbol{\psi}^*(\boldsymbol{\theta}))}{\partial \boldsymbol{\theta}} = -\mathbb{E}_{r_{\boldsymbol{\psi}^*(\boldsymbol{\theta})}(\boldsymbol{x})}[\nabla_{\boldsymbol{\theta}} E_{\boldsymbol{\theta}}(\boldsymbol{x})]$$

$\square$

## B.2 Proof for Theorem 3

**Theorem 3** (Consistent Gradient Estimation). *Let $\boldsymbol{x}_1^+, \ldots, \boldsymbol{x}_N^+$ be i.i.d. samples from $p(\boldsymbol{x})$ and $\boldsymbol{x}_1^-, \ldots, \boldsymbol{x}_N^-$ be i.i.d. samples from $q_{\boldsymbol{\theta}}(\boldsymbol{x}) \propto \exp(-E_{\boldsymbol{\theta}}(\boldsymbol{x}))$. Define the gradient estimator as:*

$$\nabla_{\boldsymbol{\theta}} \widehat{\mathcal{L}_{\gamma}^N}(\boldsymbol{\theta}; p) = -\nabla_{\boldsymbol{\theta}} \frac{1}{\gamma} \log\left(\frac{1}{N}\sum_{i=1}^{N} \exp(-\gamma E_{\boldsymbol{\theta}}(\boldsymbol{x}_i^+))\right) - \frac{\sum_{i=1}^{N} \omega_{\boldsymbol{\theta}}(\boldsymbol{x}_i^-)\nabla_{\boldsymbol{\theta}} E_{\boldsymbol{\theta}}(\boldsymbol{x}_i^-)}{\sum_{i=1}^{N} \omega_{\boldsymbol{\theta}}(\boldsymbol{x}_i^-)} \quad (22)$$

*where the self-normalized importance weight $\omega_{\boldsymbol{\theta}}(\boldsymbol{x}_i^-) := \overline{r}_{\boldsymbol{\theta}}(\boldsymbol{x}_i^-)/\overline{q}_{\boldsymbol{\theta}}(\boldsymbol{x}_i^-) = \exp(-\gamma E_{\boldsymbol{\theta}}(\boldsymbol{x}_i^-))$. Then the gradient estimator converges to the true gradient in probability:*

$$\forall \epsilon > 0, \lim_{N \to \infty} \mathbb{P}\left(\left\|\nabla_{\boldsymbol{\theta}} \widehat{\mathcal{L}_{\gamma}^N}(\boldsymbol{\theta}; p) - \nabla_{\boldsymbol{\theta}} \mathcal{L}_{\gamma}(\boldsymbol{\theta}; p)\right\| \geq \epsilon\right) = 0$$

*Proof.* First, let us write $\nabla_{\boldsymbol{\theta}} \widehat{\mathcal{L}_{\gamma}^N}(\boldsymbol{\theta}; p)$ and $\nabla_{\boldsymbol{\theta}} \mathcal{L}_{\gamma}(\boldsymbol{\theta}; p)$ as:

$$\nabla_{\boldsymbol{\theta}} \mathcal{L}_{\gamma}(\boldsymbol{\theta}; p) = \frac{\mathbb{E}_{p(\boldsymbol{x})}[\exp(-\gamma E_{\boldsymbol{\theta}}(\boldsymbol{x}))\nabla_{\boldsymbol{\theta}} E_{\boldsymbol{\theta}}(\boldsymbol{x})]}{\mathbb{E}_{p(\boldsymbol{x})}[\exp(-\gamma E_{\boldsymbol{\theta}}(\boldsymbol{x}))]} - \mathbb{E}_{r_{\boldsymbol{\theta}}(\boldsymbol{x})}[\nabla_{\boldsymbol{\theta}} E_{\boldsymbol{\theta}}(\boldsymbol{x})] \quad (23)$$

$$\nabla_{\boldsymbol{\theta}} \widehat{\mathcal{L}_{\gamma}^N}(\boldsymbol{\theta}; p) = \frac{\frac{1}{N}\sum_{i=1}^{N} \exp(-\gamma E_{\boldsymbol{\theta}}(\boldsymbol{x}_i^+))\nabla_{\boldsymbol{\theta}} E_{\boldsymbol{\theta}}(\boldsymbol{x}_i^+)}{\frac{1}{N}\sum_{i=1}^{N} \exp(-\gamma E_{\boldsymbol{\theta}}(\boldsymbol{x}_i^+))} - \frac{\frac{1}{N}\sum_{i=1}^{N} \exp(-\gamma E_{\boldsymbol{\theta}}(\boldsymbol{x}_i^-))\nabla_{\boldsymbol{\theta}} E_{\boldsymbol{\theta}}(\boldsymbol{x}_i^-)}{\frac{1}{N}\sum_{i=1}^{N} \exp(-\gamma E_{\boldsymbol{\theta}}(\boldsymbol{x}_i^-))}$$

$$(24)$$

For the first term in Equation (24), since $\{\boldsymbol{x}_i^+\}_{i=1}^N$ are *i.i.d.* samples from $p(\boldsymbol{x})$, by weak law of large numbers, the numerator and denominator of the first term in Equation (24) converges to the numerator and denominator of the first term in Equation (23) in probability. By Slutsky's theorem (*i.e.*, for random variables $X_N, X, Y_N, Y$, if $X_N \xrightarrow{p} X, Y_N \xrightarrow{p} Y$ and $X$, $Y$ are constants, then $X_N/Y_N \xrightarrow{p} X/Y$), the first term of Equation (24) converges to the first term of Equation (23) in probability.

Let us use $Z_r$ and $Z_q$ to denote the partition function for $r_{\boldsymbol{\theta}}$ and $q_{\boldsymbol{\theta}}$. The second term of Equation (24) can be written as:

$$\frac{\frac{1}{N}\sum_{i=1}^N \exp(-\gamma E_{\boldsymbol{\theta}}(\boldsymbol{x}_i^-))\nabla_{\boldsymbol{\theta}} E_{\boldsymbol{\theta}}(\boldsymbol{x}_i^-)}{\frac{1}{N}\sum_{i=1}^N \exp(-\gamma E_{\boldsymbol{\theta}}(\boldsymbol{x}_i^-))} = \frac{\frac{1}{N}\sum_{i=1}^N \frac{\exp(-(\gamma+1)E_{\boldsymbol{\theta}}(\boldsymbol{x}_i^-)/Z_r}{\exp(-E_{\boldsymbol{\theta}}(\boldsymbol{x}_i^-))/Z_q}\nabla_{\boldsymbol{\theta}} E_{\boldsymbol{\theta}}(\boldsymbol{x}_i^-)}{\frac{1}{N}\sum_{i=1}^N \frac{\exp(-(\gamma+1)E_{\boldsymbol{\theta}}(\boldsymbol{x}_i^-)/Z_r}{\exp(-E_{\boldsymbol{\theta}}(\boldsymbol{x}_i^-))/Z_q}} \tag{25}$$

Since $\{\boldsymbol{x}_i^-\}_{i=1}^N$ are *i.i.d.* samples from $q_{\boldsymbol{\theta}}(\boldsymbol{x})$, the numerator of Equation (25) converges to $\mathbb{E}_{r_{\boldsymbol{\theta}}(\boldsymbol{x})}[\nabla_{\boldsymbol{\theta}} E_{\boldsymbol{\theta}}(\boldsymbol{x})]$ in probability, while the denominator of Equation (25) converges to 1 in probability ($\mathbb{E}_{q_{\boldsymbol{\theta}}(\boldsymbol{x})}[r_{\boldsymbol{\theta}}(\boldsymbol{x})/q_{\boldsymbol{\theta}}(\boldsymbol{x})] = 1$). By Slutsky's theorem, the second term of Equation (24) converges to the second term of Equation (23) in probability. Furthermore, since convergence in probability is also preserved under addition transformation, the gradient estimator in Equation (24) converges to the true gradient in Equation (23) in probability. $\square$

## B.3 Connections to Maximum Likelihood Estimation and Extension to $\gamma < 0$

**Lemma 2.** *Let $D_\gamma(p,q)$ be the divergence corresponding to $\gamma$-scoring rule. Then, we have:*

$$\lim_{\gamma \to 0} D_\gamma(p,q) = D_{\mathrm{KL}}(p\|q)$$

*Proof.* As introduced in Equation (9) in Section 2.3, the divergence corresponding to the $\gamma$-scoring rule is:

$$
\begin{aligned}
D_\gamma(p,q) &= S_\gamma(p,p) - S_\gamma(p,q) \\
&= \frac{1}{\gamma} \log\left(\mathbb{E}_{p(\boldsymbol{x})}[p(\boldsymbol{x})^\gamma]\right) - \log(\|p\|_{\gamma+1}) - \frac{1}{\gamma} \log\left(\mathbb{E}_{p(\boldsymbol{x})}[q(\boldsymbol{x})^\gamma]\right) + \log(\|q\|_{\gamma+1}) \\
&= -\frac{1}{\gamma} \log\left(\int_{\mathcal{X}} p(\boldsymbol{x})q(\boldsymbol{x})^\gamma \mathrm{d}\boldsymbol{x}\right) + \frac{1}{\gamma+1} \log\left(\int_{\mathcal{X}} q(\boldsymbol{x})^{\gamma+1} \mathrm{d}\boldsymbol{x}\right) + \frac{1}{\gamma(\gamma+1)} \log\left(\int_{\mathcal{X}} p(\boldsymbol{x})^{\gamma+1} \mathrm{d}\boldsymbol{x}\right)
\end{aligned}
$$

When $\gamma \to 0$, with Taylor series, we know that:

$$
\begin{aligned}
q^\gamma &= 1 + \gamma \log(q) + \mathcal{O}(\gamma^2) \\
p^\gamma &= 1 + \gamma \log(p) + \mathcal{O}(\gamma^2)
\end{aligned}
$$

Therefore, we have:

$$
\begin{aligned}
\lim_{\gamma \to 0} D_\gamma(p,q) &= \lim_{\gamma \to 0} -\frac{1}{\gamma} \log\left(\int_{\mathcal{X}} p(\boldsymbol{x})(1 + \gamma \log q(\boldsymbol{x}) + \mathcal{O}(\gamma^2))\mathrm{d}\boldsymbol{x}\right) \\
&\quad + \frac{1}{\gamma+1} \log\left(\int_{\mathcal{X}} q(\boldsymbol{x})(1 + \gamma \log q(\boldsymbol{x}) + \mathcal{O}(\gamma^2))\mathrm{d}\boldsymbol{x}\right) \\
&\quad + \frac{1}{\gamma(\gamma+1)} \log\left(\int_{\mathcal{X}} p(\boldsymbol{x})(1 + \gamma \log p(\boldsymbol{x}) + \mathcal{O}(\gamma^2))\mathrm{d}\boldsymbol{x}\right) \\
&= \lim_{\gamma \to 0} -\frac{1}{\gamma} \log\left(1 + \gamma \int_{\mathcal{X}} p(\boldsymbol{x}) \log q(\boldsymbol{x})\mathrm{d}\boldsymbol{x} + \mathcal{O}(\gamma^2)\right) \\
&\quad + \frac{1}{\gamma+1} \log\left(1 + \gamma \int_{\mathcal{X}} q(\boldsymbol{x}) \log q(\boldsymbol{x})\mathrm{d}\boldsymbol{x} + \mathcal{O}(\gamma^2)\right) \\
&\quad + \frac{1}{\gamma(\gamma+1)} \log\left(1 + \gamma \int_{\mathcal{X}} p(\boldsymbol{x}) \log p(\boldsymbol{x})\mathrm{d}\boldsymbol{x} + \mathcal{O}(\gamma^2))\right) \\
&= \lim_{\gamma \to 0} -\int_{\mathcal{X}} p(\boldsymbol{x}) \log q(\boldsymbol{x})\mathrm{d}\boldsymbol{x} + \frac{1}{\gamma+1} \int_{\mathcal{X}} p(\boldsymbol{x}) \log p(\boldsymbol{x})\mathrm{d}\boldsymbol{x} + \mathcal{O}(\gamma) \\
&= \int_{\mathcal{X}} p(\boldsymbol{x}) \log \frac{p(\boldsymbol{x})}{q(\boldsymbol{x})} = D_{\mathrm{KL}}(p\|q)
\end{aligned}
$$

$\square$

The above lemma implies that the KL divergence minimization (maximum likelihood estimation) is a special case of $\gamma$-divergence minimization when $\gamma \to 0$, which also implies the following corollary:

**Corollary 1.** *When $\gamma \to 0$, the gradient of pseudo-spherical contrastive divergence is equal to the gradient of contrastive divergence:*

$$\lim_{\gamma \to 0} \nabla_{\boldsymbol{\theta}} \mathcal{L}_\gamma(\boldsymbol{\theta}; p) = \nabla_{\boldsymbol{\theta}} \mathcal{L}_{\mathrm{MLE}}(\boldsymbol{\theta}; p)$$

*Proof.* This is a direct consequence of Lemma 2. It can also be verified by checking the PS-CD gradient in Equation (18) (when $\gamma \to 0$, $r_{\boldsymbol{\theta}} = q_{\boldsymbol{\theta}} \propto \exp(-E_{\boldsymbol{\theta}})$):

$$
\begin{aligned}
\lim_{\gamma \to 0} \nabla_{\boldsymbol{\theta}} \mathcal{L}_\gamma(\boldsymbol{\theta}; p) &= \lim_{\gamma \to 0} -\frac{1}{\gamma} \nabla_{\boldsymbol{\theta}} \log\left(\mathbb{E}_{p(\boldsymbol{x})}[\exp(-\gamma E_{\boldsymbol{\theta}}(\boldsymbol{x}))]\right) - \mathbb{E}_{r_{\boldsymbol{\theta}}(\boldsymbol{x})}[\nabla_{\boldsymbol{\theta}} E_{\boldsymbol{\theta}}(\boldsymbol{x})] \\
&= \lim_{\gamma \to 0} -\frac{1}{\gamma} \frac{-\gamma \mathbb{E}_{p(\boldsymbol{x})}[\exp(-\gamma E_{\boldsymbol{\theta}}(\boldsymbol{x}))\nabla_{\boldsymbol{\theta}} E_{\boldsymbol{\theta}}(\boldsymbol{x})]}{\mathbb{E}_{p(\boldsymbol{x})}[\exp(-\gamma E_{\boldsymbol{\theta}}(\boldsymbol{x}))]} - \mathbb{E}_{r_{\boldsymbol{\theta}}(\boldsymbol{x})}[\nabla_{\boldsymbol{\theta}} E_{\boldsymbol{\theta}}(\boldsymbol{x})] \\
&= \mathbb{E}_{p(\boldsymbol{x})}[\nabla_{\boldsymbol{\theta}} E_{\boldsymbol{\theta}}(\boldsymbol{x})] - \mathbb{E}_{q_{\boldsymbol{\theta}}(\boldsymbol{x})}[\nabla_{\boldsymbol{\theta}} E_{\boldsymbol{\theta}}(\boldsymbol{x})] = \nabla_{\boldsymbol{\theta}} \mathcal{L}_{\mathrm{MLE}}(\boldsymbol{\theta}; p)
\end{aligned}
$$

$\square$

Inspired by [86, 56] that generalize Rényi divergence beyond its definition to negative orders, we consider the extension of $\gamma$-scoring rule with $\gamma < 0$ (although it is no longer strictly proper for these $\gamma$ values) and show that maximizing such scoring rule is equivalent to maximizing a lower bound of logarithm scoring rule (MLE) with an additional Rényi entropy regularization.

**Lemma 3.** *When $-1 \leq \gamma < 0$, we have:*

$$S_\gamma(p, q) \leq \mathbb{E}_{p(\boldsymbol{x})}[\log q(\boldsymbol{x})] + \frac{\gamma}{\gamma + 1} \mathcal{H}_{\gamma+1}(q)$$

*where $\mathcal{H}_{\gamma+1}(q)$ is the Rényi entropy of order $\gamma + 1$.*

*Proof.* As a generalization to Shannon entropy, the Rényi entropy of order $\alpha$ is defined as:

$$\mathcal{H}_\alpha(q) = \frac{\alpha}{1 - \alpha} \log(\|q\|_\alpha)$$

With Jensen's inequality, for $-1 \leq \gamma < 0$, we have:

$$
\begin{aligned}
S_\gamma(p, q) &= \frac{1}{\gamma} \log(\mathbb{E}_{p(\boldsymbol{x})}[q(\boldsymbol{x})^\gamma]) - \log(\|q\|_{\gamma+1}) \\
&\leq \frac{1}{\gamma} \mathbb{E}_{p(\boldsymbol{x})}[\gamma \log(q(\boldsymbol{x}))] + \frac{\gamma}{\gamma + 1} \left( \frac{\gamma + 1}{-\gamma} \log(\|q\|_{\gamma+1}) \right) \\
&= \mathbb{E}_{p(\boldsymbol{x})}[\log q(\boldsymbol{x})] + \frac{\gamma}{\gamma + 1} \mathcal{H}_{\gamma+1}(q)
\end{aligned}
$$

$\square$

## C  Theoretical Analysis

In this section, we provide a theoretical analysis on the sample complexity of the gradient estimator, as well as the convergence property of stochastic gradient descent with consistent (but biased given finite samples) gradient estimators as presented in Algorithm 1.

### C.1  Sample Complexity

We start with analyzing the sample complexity of the consistent gradient estimator, that is how fast it approaches the true gradient value or how many samples we need in order to empirically estimate the gradient at a given accuracy with a high probability.

We first make the following assumption, which is similar to the one used in [4, 47]:

**Assumption 1.** *The energy function is bounded by $K$ and the gradient is bounded by $L$ (with $K > 0$ and $L > 0$):*
$$\forall \boldsymbol{x} \in \mathcal{X}, \ \boldsymbol{\theta} \in \Theta, \ |E_{\boldsymbol{\theta}}(\boldsymbol{x})| \leq K, \ \|\nabla_{\boldsymbol{\theta}} E_{\boldsymbol{\theta}}(\boldsymbol{x})\| \leq L.$$

The assumption is typically easy to enforce in practice. For example, in the experiments we use $L_2$ regularization on the outputs of the energy function, as well as normalized inputs and spectral normalization [60] for the neural network that realizes the energy function.

**Theorem 4.** *Under Assumption 1, given any constants $\epsilon > 0$ and $\delta \in (0, 1)$, when the number of samples $N$ satisfies:*

$$N \geq \frac{32 L^2 e^{8\gamma K} (1 + 4 \log(2/\delta))}{\epsilon^2}$$

*we have:*

$$\mathbb{P}\left( \left\| \nabla_{\boldsymbol{\theta}} \widehat{\mathcal{L}_\gamma^N}(\boldsymbol{\theta}; p) - \nabla_{\boldsymbol{\theta}} \mathcal{L}_\gamma(\boldsymbol{\theta}; p) \right\| \leq \epsilon \right) \geq 1 - \delta$$

*Proof.* For notation simplicity, we use $p^N$ to denote the empirical distribution of $\{\boldsymbol{x}_i\}_{i=1}^N$ *i.i.d.* sampled from a distribution $p$, *i.e.*, $\mathbb{E}_{p^N(\boldsymbol{x})}[f(\boldsymbol{x})] = \frac{1}{N} \sum_{i=1}^N f(\boldsymbol{x}_i)$. Similarly, $\mathbb{E}_{q_{\boldsymbol{\theta}}^N(\boldsymbol{x})}[f(\boldsymbol{x})] = \sum_{i=1}^N f(\boldsymbol{x}_i)$ when $\{\boldsymbol{x}_i\}_{i=1}^N$ are *i.i.d.* samples from $q_{\boldsymbol{\theta}}$.

First, we observe that:

$$\mathbb{E}_{r_{\boldsymbol{\theta}}(\boldsymbol{x})}[\nabla_{\boldsymbol{\theta}} E_{\boldsymbol{\theta}}(\boldsymbol{x})] = \frac{\mathbb{E}_{q_{\boldsymbol{\theta}}(\boldsymbol{x})}\left[\frac{r_{\boldsymbol{\theta}}(\boldsymbol{x})}{q_{\boldsymbol{\theta}}(\boldsymbol{x})}\nabla_{\boldsymbol{\theta}} E_{\boldsymbol{\theta}}(\boldsymbol{x})\right]}{\mathbb{E}_{q_{\boldsymbol{\theta}}(\boldsymbol{x})}\left[\frac{r_{\boldsymbol{\theta}}(\boldsymbol{x})}{q_{\boldsymbol{\theta}}(\boldsymbol{x})}\right]} = \frac{\mathbb{E}_{q_{\boldsymbol{\theta}}(\boldsymbol{x})}[\exp(-\gamma E_{\boldsymbol{\theta}}(\boldsymbol{x}))\nabla_{\boldsymbol{\theta}} E_{\boldsymbol{\theta}}(\boldsymbol{x})]}{\mathbb{E}_{q_{\boldsymbol{\theta}}(\boldsymbol{x})}[\exp(-\gamma E_{\boldsymbol{\theta}}(\boldsymbol{x}))]}$$

where the partition functions of $r_{\boldsymbol{\theta}}$ and $q_{\boldsymbol{\theta}}$ cancel out. Based on Equation (23) and (24), with triangle inequality, the estimation error can be upper bounded as:

$$
\begin{aligned}
&\left\|\nabla_{\boldsymbol{\theta}}\widehat{\mathcal{L}_{\gamma}^{N}(\boldsymbol{\theta}; p)} - \nabla_{\boldsymbol{\theta}}\mathcal{L}_{\gamma}(\boldsymbol{\theta}; p)\right\| \\
&\leq \underbrace{\left\|\frac{\mathbb{E}_{p(\boldsymbol{x})}[\exp(-\gamma E_{\boldsymbol{\theta}}(\boldsymbol{x}))\nabla_{\boldsymbol{\theta}} E_{\boldsymbol{\theta}}(\boldsymbol{x})]}{\mathbb{E}_{p(\boldsymbol{x})}[\exp(-\gamma E_{\boldsymbol{\theta}}(\boldsymbol{x}))]} - \frac{\mathbb{E}_{p^{N}(\boldsymbol{x})}[\exp(-\gamma E_{\boldsymbol{\theta}}(\boldsymbol{x}))\nabla_{\boldsymbol{\theta}} E_{\boldsymbol{\theta}}(\boldsymbol{x})]}{\mathbb{E}_{p^{N}(\boldsymbol{x})}[\exp(-\gamma E_{\boldsymbol{\theta}}(\boldsymbol{x}))]}\right\|}_{\Delta_p} + \\
&\quad \underbrace{\left\|\frac{\mathbb{E}_{q_{\boldsymbol{\theta}}(\boldsymbol{x})}[\exp(-\gamma E_{\boldsymbol{\theta}}(\boldsymbol{x}))\nabla_{\boldsymbol{\theta}} E_{\boldsymbol{\theta}}(\boldsymbol{x})]}{\mathbb{E}_{q_{\boldsymbol{\theta}}(\boldsymbol{x})}[\exp(-\gamma E_{\boldsymbol{\theta}}(\boldsymbol{x}))]} - \frac{\mathbb{E}_{q_{\boldsymbol{\theta}}^{N}(\boldsymbol{x})}[\exp(-\gamma E_{\boldsymbol{\theta}}(\boldsymbol{x}))\nabla_{\boldsymbol{\theta}} E_{\boldsymbol{\theta}}(\boldsymbol{x})]}{\mathbb{E}_{q_{\boldsymbol{\theta}}^{N}(\boldsymbol{x})}[\exp(-\gamma E_{\boldsymbol{\theta}}(\boldsymbol{x}))]}\right\|}_{\Delta_{q_{\boldsymbol{\theta}}}}
\end{aligned}
\tag{26}
$$

Define functions:

$$f_{\boldsymbol{\theta}}(\boldsymbol{x}) := \exp(-\gamma E_{\boldsymbol{\theta}}(\boldsymbol{x})), \quad \boldsymbol{h}_{\boldsymbol{\theta}}(\boldsymbol{x}) := \exp(-\gamma E_{\boldsymbol{\theta}}(\boldsymbol{x}))\nabla_{\boldsymbol{\theta}} E_{\boldsymbol{\theta}}(\boldsymbol{x})$$

From Assumption 1, we know that:

$$\forall \boldsymbol{x} \in \mathcal{X}, \boldsymbol{\theta} \in \Theta, f_{\boldsymbol{\theta}}(\boldsymbol{x}) \in [e^{-\gamma K}, e^{\gamma K}], \|\boldsymbol{h}_{\boldsymbol{\theta}}(\boldsymbol{x})\| \leq Le^{\gamma K} \tag{27}$$

Let us examine the first term $\Delta_p$ in Equation (26):

$$
\begin{aligned}
\Delta_p &= \left\|\frac{\mathbb{E}_{p(\boldsymbol{x})}[\boldsymbol{h}_{\boldsymbol{\theta}}(\boldsymbol{x})]}{\mathbb{E}_{p(\boldsymbol{x})}[f_{\boldsymbol{\theta}}(\boldsymbol{x})]} - \frac{\mathbb{E}_{p^{N}(\boldsymbol{x})}[\boldsymbol{h}_{\boldsymbol{\theta}}(\boldsymbol{x})]}{\mathbb{E}_{p^{N}(\boldsymbol{x})}[f_{\boldsymbol{\theta}}(\boldsymbol{x})]}\right\| \\
&= \frac{1}{\mathbb{E}_{p(\boldsymbol{x})}[f_{\boldsymbol{\theta}}(\boldsymbol{x})] \cdot \mathbb{E}_{p^{N}(\boldsymbol{x})}[f_{\boldsymbol{\theta}}(\boldsymbol{x})]}\left\|\mathbb{E}_{p^{N}(\boldsymbol{x})}[f_{\boldsymbol{\theta}}(\boldsymbol{x})] \cdot \mathbb{E}_{p(\boldsymbol{x})}[\boldsymbol{h}_{\boldsymbol{\theta}}(\boldsymbol{x})] - \mathbb{E}_{p(\boldsymbol{x})}[f_{\boldsymbol{\theta}}(\boldsymbol{x})] \cdot \mathbb{E}_{p^{N}(\boldsymbol{x})}[\boldsymbol{h}_{\boldsymbol{\theta}}(\boldsymbol{x})]\right\| \\
&\leq e^{2\gamma K}\left\|\mathbb{E}_{p^{N}(\boldsymbol{x})}[f_{\boldsymbol{\theta}}(\boldsymbol{x})] \cdot \mathbb{E}_{p(\boldsymbol{x})}[\boldsymbol{h}_{\boldsymbol{\theta}}(\boldsymbol{x})] - \mathbb{E}_{p(\boldsymbol{x})}[f_{\boldsymbol{\theta}}(\boldsymbol{x})] \cdot \mathbb{E}_{p^{N}(\boldsymbol{x})}[\boldsymbol{h}_{\boldsymbol{\theta}}(\boldsymbol{x})]\right\|
\end{aligned}
\tag{28}
$$

Now we introduce the following lemma that will provide us a probability upper bound that an empirical mean of independent random variables deviates from its expected value more than a certain amount.

**Lemma 4** (Vector Bernstein Inequality [47, 32]). *Let $\boldsymbol{X}_1, \ldots, \boldsymbol{X}_N$ be independent vector-valued random variables. Assume that each one is centered, uniformly bounded and the variance is also bounded:*

$$\forall i, \mathbb{E}[\boldsymbol{X}_i] = 0 \text{ and } \|\boldsymbol{X}_i\| \leq \mu \text{ and } \mathbb{E}[\|\boldsymbol{X}_i\|^2] \leq \sigma^2$$

*Define $\overline{\boldsymbol{X}} := \frac{1}{N}(\boldsymbol{X}_1 + \ldots + \boldsymbol{X}_N)$. Then we have for $0 < t < \sigma^2/\mu$:*

$$\mathbb{P}(\|\overline{\boldsymbol{X}}\| \geq t) \leq \exp\left(-\frac{Nt^2}{8\sigma^2} + \frac{1}{4}\right)$$

We then define the following vector-valued random variable:

$$
\begin{aligned}
\boldsymbol{X}_i &:= f_{\boldsymbol{\theta}}(\boldsymbol{x}_i) \cdot \mathbb{E}_{p(\boldsymbol{x})}[\boldsymbol{h}_{\boldsymbol{\theta}}(\boldsymbol{x})] - \mathbb{E}_{p(\boldsymbol{x})}[f_{\boldsymbol{\theta}}(\boldsymbol{x})] \cdot \boldsymbol{h}_{\boldsymbol{\theta}}(\boldsymbol{x}_i) \\
\overline{\boldsymbol{X}} &:= \frac{1}{N}\sum_{i=1}^{N} f_{\boldsymbol{\theta}}(\boldsymbol{x}_i) \cdot \mathbb{E}_{p(\boldsymbol{x})}[\boldsymbol{h}_{\boldsymbol{\theta}}(\boldsymbol{x})] - \mathbb{E}_{p(\boldsymbol{x})}[f_{\boldsymbol{\theta}}(\boldsymbol{x})] \cdot \frac{1}{N}\sum_{i=1}^{N}\boldsymbol{h}_{\boldsymbol{\theta}}(\boldsymbol{x}_i) \\
&= \mathbb{E}_{p^{N}(\boldsymbol{x})}[f_{\boldsymbol{\theta}}(\boldsymbol{x})] \cdot \mathbb{E}_{p(\boldsymbol{x})}[\boldsymbol{h}_{\boldsymbol{\theta}}(\boldsymbol{x})] - \mathbb{E}_{p(\boldsymbol{x})}[f_{\boldsymbol{\theta}}(\boldsymbol{x})] \cdot \mathbb{E}_{p^{N}(\boldsymbol{x})}[\boldsymbol{h}_{\boldsymbol{\theta}}(\boldsymbol{x})]
\end{aligned}
$$

From Equation (27), we know that:

$$
\begin{aligned}
\|\boldsymbol{X}_i\| &= \|f_{\boldsymbol{\theta}}(\boldsymbol{x}_i) \cdot \mathbb{E}_{p(\boldsymbol{x})}[\boldsymbol{h}_{\boldsymbol{\theta}}(\boldsymbol{x})] - \mathbb{E}_{p(\boldsymbol{x})}[f_{\boldsymbol{\theta}}(\boldsymbol{x})] \cdot \boldsymbol{h}_{\boldsymbol{\theta}}(\boldsymbol{x}_i)\| \\
&\leq \|f_{\boldsymbol{\theta}}(\boldsymbol{x}_i) \cdot \mathbb{E}_{p(\boldsymbol{x})}[\boldsymbol{h}_{\boldsymbol{\theta}}(\boldsymbol{x})]\| + \|\mathbb{E}_{p(\boldsymbol{x})}[f_{\boldsymbol{\theta}}(\boldsymbol{x})] \cdot \boldsymbol{h}_{\boldsymbol{\theta}}(\boldsymbol{x}_i)\| \leq 2Le^{2\gamma K} \\
\|\boldsymbol{X}_i\|^2 &\leq 4L^2 e^{4\gamma K}
\end{aligned}
$$

With $\sigma^2 := 4L^2 e^{4\gamma K}$, from Lemma 4, we know that:

$$\mathbb{P}\left(e^{2\gamma K}\left\|\mathbb{E}_{p^N(\boldsymbol{x})}[f_{\boldsymbol{\theta}}(\boldsymbol{x})]\cdot\mathbb{E}_{p(\boldsymbol{x})}[h_{\boldsymbol{\theta}}(\boldsymbol{x})]-\mathbb{E}_{p(\boldsymbol{x})}[f_{\boldsymbol{\theta}}(\boldsymbol{x})]\cdot\mathbb{E}_{p^N(\boldsymbol{x})}[h_{\boldsymbol{\theta}}(\boldsymbol{x})]\right\|\geq\frac{\epsilon}{2}\right)$$
$$\leq\exp\left(-\frac{N\epsilon^2}{128L^2 e^{8\gamma K}}+\frac{1}{4}\right) \tag{29}$$

To obtain a sample complexity bound such that the probability bound in Equation (29) is less than $1-\sqrt{1-\delta}$, we need to solve for $N$:

$$\exp\left(-\frac{N\epsilon^2}{128L^2 e^{8\gamma K}}+\frac{1}{4}\right)\leq 1-\sqrt{1-\delta} \tag{30}$$

Solving Equation (30) gives us:

$$N\geq\frac{32L^2 e^{8\gamma K}\left(1-4\log\left(1-\sqrt{1-\delta}\right)\right)}{\epsilon^2} \tag{31}$$

Because $1+4\log(2/\delta)>1-4\log(1-\sqrt{1-\delta})$ for $\delta\in(0,1]$, we use the following slightly weaker bound such that it looks cleaner:

$$N\geq\frac{32L^2 e^{8\gamma K}\left(1+4\log(2/\delta)\right)}{\epsilon^2} \tag{32}$$

Since Equation (28) is an upper bound of $\Delta_p$, we know that when the sample size satisfies Equation (32), we have:

$$\mathbb{P}\left(\Delta_p=\left\|\frac{\mathbb{E}_{p(\boldsymbol{x})}[h_{\boldsymbol{\theta}}(\boldsymbol{x})]}{\mathbb{E}_{p(\boldsymbol{x})}[f_{\boldsymbol{\theta}}(\boldsymbol{x})]}-\frac{\mathbb{E}_{p^N(\boldsymbol{x})}[h_{\boldsymbol{\theta}}(\boldsymbol{x})]}{\mathbb{E}_{p^N(\boldsymbol{x})}[f_{\boldsymbol{\theta}}(\boldsymbol{x})]}\right\|\leq\frac{\epsilon}{2}\right)$$
$$\geq\mathbb{P}\left(e^{2\gamma K}\left\|\mathbb{E}_{p^N(\boldsymbol{x})}[f_{\boldsymbol{\theta}}(\boldsymbol{x})]\cdot\mathbb{E}_{p(\boldsymbol{x})}[h_{\boldsymbol{\theta}}(\boldsymbol{x})]-\mathbb{E}_{p(\boldsymbol{x})}[f_{\boldsymbol{\theta}}(\boldsymbol{x})]\cdot\mathbb{E}_{p^N(\boldsymbol{x})}[h_{\boldsymbol{\theta}}(\boldsymbol{x})]\right\|\leq\frac{\epsilon}{2}\right)$$
$$\geq\sqrt{1-\delta}$$

Similarly, we can obtain the same sample complexity bound for $\Delta_{q_{\boldsymbol{\theta}}}$ such that:

$$\mathbb{P}\left(\Delta_{q_{\boldsymbol{\theta}}}=\left\|\frac{\mathbb{E}_{q_{\boldsymbol{\theta}}(\boldsymbol{x})}[h_{\boldsymbol{\theta}}(\boldsymbol{x})]}{\mathbb{E}_{q_{\boldsymbol{\theta}}(\boldsymbol{x})}[f_{\boldsymbol{\theta}}(\boldsymbol{x})]}-\frac{\mathbb{E}_{q_{\boldsymbol{\theta}}^N(\boldsymbol{x})}[h_{\boldsymbol{\theta}}(\boldsymbol{x})]}{\mathbb{E}_{q_{\boldsymbol{\theta}}^N(\boldsymbol{x})}[f_{\boldsymbol{\theta}}(\boldsymbol{x})]}\right\|\leq\frac{\epsilon}{2}\right)\geq\sqrt{1-\delta} \tag{33}$$

From Equation (26), we know that $\Delta_p+\Delta_{q_{\boldsymbol{\theta}}}$ is an upper bound of the gradient estimation error. Also note that the event $\Delta_p\leq\frac{\epsilon}{2}$ and the event $\Delta_{q_{\boldsymbol{\theta}}}\leq\frac{\epsilon}{2}$ are independent from each other (the samples for $p^N$ and the samples for $q_{\boldsymbol{\theta}}^N$ are independent samples from $p$ and $q_{\boldsymbol{\theta}}$ respectively). Thus when the sample size satisfies Equation (32), we have:

$$\mathbb{P}\left(\|\nabla_{\boldsymbol{\theta}}\widehat{\mathcal{L}_\gamma^N}(\boldsymbol{\theta};p)-\nabla_{\boldsymbol{\theta}}\mathcal{L}_\gamma(\boldsymbol{\theta};p)\|\leq\epsilon\right)$$
$$\geq\mathbb{P}\left(\Delta_p+\Delta_{q_{\boldsymbol{\theta}}}\leq\epsilon\right)$$
$$\geq\mathbb{P}\left(\Delta_p\leq\frac{\epsilon}{2}\text{ and }\Delta_{q_{\boldsymbol{\theta}}}\leq\frac{\epsilon}{2}\right)$$
$$=\mathbb{P}\left(\Delta_p\leq\frac{\epsilon}{2}\right)\cdot\mathbb{P}\left(\Delta_{q_{\boldsymbol{\theta}}}\leq\frac{\epsilon}{2}\right)$$
$$\geq 1-\delta$$

□

## C.2 Convergence of Pseudo-Spherical Contrastive Divergence Algorithm

In this section, we analyze the convergence property of the PS-CD algorithm presented in Algorithm 1. For notation simplicity, we define $\boldsymbol{g}$ as the true gradient in Equation (18) and $\widehat{\boldsymbol{g}}$ as the gradient estimator in Equation (19). We further use $\mathcal{L}(\boldsymbol{\theta})$ to denote the loss function $\mathcal{L}_\gamma(\boldsymbol{\theta},\psi^*(\boldsymbol{\theta});p)=-S_\gamma(p,q_{\boldsymbol{\theta}})$.

Let us consider the following stochastic gradient descent (SGD) update rule:

$$\boldsymbol{\theta}_{t+1} = \boldsymbol{\theta}_t - \eta_t \widehat{\boldsymbol{g}}_t, \ \ t = 1, 2, \ldots, T \tag{34}$$

where $\eta_t$ is the step size at step $t$, $T$ is the total number of steps and $\widehat{\boldsymbol{g}}_t := \nabla_{\boldsymbol{\theta}} \widehat{\mathcal{L}_\gamma^N(\boldsymbol{\theta}; p)}\big|_{\boldsymbol{\theta}=\boldsymbol{\theta}_t}$ is the consistent (but biased) gradient estimation of $\boldsymbol{g}_t$ at step $t$. Note that $\boldsymbol{\theta}_t$ and $\widehat{\boldsymbol{g}}_t$ are random variables that depend on the previous history $\widehat{\boldsymbol{g}}_1, \ldots, \widehat{\boldsymbol{g}}_{t-1}$. For brevity, in the following we will omit such dependency in the notations.

Most works for analyzing the convergence behavior of SGD relies on the assumption that the gradient estimator $\widehat{\boldsymbol{g}}_t$ is asymptotically unbiased, *e.g.*, [63, 52, 78, 24, 74], while in our case the gradient estimator is not unbiased but consistent (see Section 1.2 in [8] for a detailed discussion on the distinctions between unbiasedness and consistency). Therefore, in this work we generalize the theory developed in [24] and [8] to analyze the convergence rate for PS-CD.

Besides Assumption 1 used for analyzing the sample complexity of the gradient estimator, we further make the following assumption:

**Assumption 2.** *The loss function $\mathcal{L}(\boldsymbol{\theta})$ is $M$-smooth (with $M > 0$):*

$$\forall \boldsymbol{\theta}_1, \boldsymbol{\theta}_2 \in \Theta, \ \|\nabla\mathcal{L}(\boldsymbol{\theta}_1) - \nabla\mathcal{L}(\boldsymbol{\theta}_2)\| \leq M\|\boldsymbol{\theta}_1 - \boldsymbol{\theta}_2\|.$$

This is a common assumption used for analyzing first-order optimization methods, which is also used in [24, 8]. Also note that this is a relatively mild assumption since we do not require the loss function to be convex in $\boldsymbol{\theta}$. Since in non-convex optimization, the convergence criterion is typically measured by gradient norm, following [64, 24], we use $\|\nabla\mathcal{L}(\boldsymbol{\theta})\| \leq \xi$ to judge whether a solution $\boldsymbol{\theta}$ is approximately a stationary point.

Now, let us consider Algorithm 2, which is a variant of SGD that allows early stopping before reaching the iteration limit $T$ according to some probability distribution $p_Z$ over iteration indexes $[T] := \{1, \ldots, T\}$.

---

**Algorithm 2** Randomized Stochastic Gradient Descent

---

1: **Input:** Initial parameter $\boldsymbol{\theta}_1$, iteration limit $T$, step sizes $\{\eta_t\}_{t=1}^T$, distribution $p_Z$ over $[T]$.
2: Sample an iteration number $Z$ from $p_Z$ (defined in Equation (35)).
3: **for** $t = 1, \ldots, Z$ **do**
4:     Obtain the gradient estimator $\widehat{\boldsymbol{g}}_t$ with a sample batch size of $N_t$.
5:     Update the parameter: $\boldsymbol{\theta}_{t+1} = \boldsymbol{\theta}_t - \eta_t \widehat{\boldsymbol{g}}_t$.
6: **end for**
7: **Output:** $\boldsymbol{\theta}_Z$.

---

Note that this is equivalent (more efficient in terms of computation) to running the algorithm to the iteration limit $T$ and then selecting the final solution from $\{\boldsymbol{\theta}_1, \ldots, \boldsymbol{\theta}_T\}$ according to distribution $p_Z$.

We have the following theorem that characterizes the convergence property of Algorithm 2:

**Theorem 5.** *Under Assumptions 1 and 2, for arbitrary constants $\alpha \in (0, 1)$ and $\delta \in (0, 1)$, suppose that the step sizes satisfy $\eta_t < 2(1 - \alpha)/M$ and the probability distribution over iteration indexes is chosen to be:*

$$p_Z(t) := \frac{2(1 - \alpha)\eta_t - M\eta_t^2}{\sum_{t=1}^T (2(1 - \alpha)\eta_t - M\eta_t^2)}, \ \ t = 1, \ldots, T \tag{35}$$

*and the sample size $N_t$ used for estimating $\widehat{\boldsymbol{g}}_t$ satisfies:*

$$N_t \geq \frac{32L^2 e^{8\gamma K}(1 + 4\log(2T/\delta))}{\alpha^2 \|\boldsymbol{g}_t\|^2} \tag{36}$$

*Denote by $\mathcal{L}^*$ the minimum value of $\mathcal{L}(\boldsymbol{\theta})$. Then with probability at least $1 - \delta$, we have:*

$$\mathbb{E}_{p_Z}[\|\nabla\mathcal{L}(\boldsymbol{\theta}_Z)\|^2] < \frac{2(\mathcal{L}(\boldsymbol{\theta}_1) - \mathcal{L}^*) + 12\alpha M L^2 e^{4\gamma K} \sum_{t=1}^T \eta_t^2}{\sum_{t=1}^T (2(1 - \alpha)\eta_t - M\eta_t^2)} \tag{37}$$

*Proof.* First, with Assumption 1 ($|E_{\boldsymbol{\theta}}(\boldsymbol{x})| \leq K$, $\|\nabla_{\boldsymbol{\theta}} E_{\boldsymbol{\theta}}(\boldsymbol{x})\| \leq L$), we can bound the norm of the true gradient as:

$$\|\boldsymbol{g}\| \leq \left\| \frac{\mathbb{E}_{p(\boldsymbol{x})}[\exp(-\gamma E_{\boldsymbol{\theta}}(\boldsymbol{x}))\nabla_{\boldsymbol{\theta}} E_{\boldsymbol{\theta}}(\boldsymbol{x})]}{\mathbb{E}_{p(\boldsymbol{x})}[\exp(-\gamma E_{\boldsymbol{\theta}}(\boldsymbol{x}))]} \right\| + \|\mathbb{E}_{r_{\psi^*(\boldsymbol{\theta})}(\boldsymbol{x})}[\nabla_{\boldsymbol{\theta}} E_{\boldsymbol{\theta}}(\boldsymbol{x})]\|$$
$$\leq Le^{2\gamma K} + L < 2Le^{2\gamma K} \tag{38}$$

From Theorem 4, we know that when sample size at each step satisfies Equation (36), we have:

$$\mathbb{P}(\|\widehat{\boldsymbol{g}}_t - \boldsymbol{g}_t\| \leq \alpha\|\boldsymbol{g}_t\|) \geq 1 - \delta/T$$

Therefore, we have:

$$\mathbb{P}(\|\widehat{\boldsymbol{g}}_1 - \boldsymbol{g}_1\| \leq \alpha\|\boldsymbol{g}_1\| \text{ and } \ldots \text{ and } \|\widehat{\boldsymbol{g}}_T - \boldsymbol{g}_T\| \leq \alpha\|\boldsymbol{g}_T\|) \geq \prod_{t=1}^{T}(1 - \delta/T) \geq 1 - \delta$$

Thus, with probability at least $1 - \delta$, we have:

$$\|\widehat{\boldsymbol{g}}_1 - \boldsymbol{g}_1\| \leq \alpha\|\boldsymbol{g}_1\| \text{ and } \ldots \text{ and } \|\widehat{\boldsymbol{g}}_T - \boldsymbol{g}_T\| \leq \alpha\|\boldsymbol{g}_T\| \tag{39}$$

A similar condition was also adopted in [38] and [8]. When Equation (39) is satisfied, we have the following lemma:

**Lemma 5** (Lemma 11 in [8]). *If $\|\widehat{\boldsymbol{g}}_t - \boldsymbol{g}_t\| \leq \alpha\|\boldsymbol{g}_t\|$, then we have:*

$$(1 - \alpha)\|\boldsymbol{g}_t\| \leq \|\widehat{\boldsymbol{g}}_t\| \leq (1 + \alpha)\|\boldsymbol{g}_t\|$$

Next we introduce the following property of $M$-smooth function:

**Lemma 6.** *For an $M$-smooth function $\mathcal{L}(\boldsymbol{\theta})$, we have:*

$$\forall \boldsymbol{\theta}_1, \boldsymbol{\theta}_2 \in \Theta, \ \mathcal{L}(\boldsymbol{\theta}_2) \leq \mathcal{L}(\boldsymbol{\theta}_1) + \langle \nabla\mathcal{L}(\boldsymbol{\theta}_1), \boldsymbol{\theta}_2 - \boldsymbol{\theta}_1 \rangle + \frac{M}{2}\|\boldsymbol{\theta}_2 - \boldsymbol{\theta}_1\|^2$$

From Assumption 2 and Lemma 6, we know that:

$$\mathcal{L}(\boldsymbol{\theta}_{t+1}) \leq \mathcal{L}(\boldsymbol{\theta}_t) + \langle \nabla\mathcal{L}(\boldsymbol{\theta}_t), \boldsymbol{\theta}_{t+1} - \boldsymbol{\theta}_t \rangle + \frac{M}{2}\|\boldsymbol{\theta}_{t+1} - \boldsymbol{\theta}_t\|^2$$

From the SGD update rule in Equation (34) ($\boldsymbol{\theta}_{t+1} = \boldsymbol{\theta}_t - \eta_t\widehat{\boldsymbol{g}}_t$), Equation (39) and Lemma 5, we know that:

$$\mathcal{L}(\boldsymbol{\theta}_{t+1}) \leq \mathcal{L}(\boldsymbol{\theta}_t) - \eta_t\langle \boldsymbol{g}_t, \widehat{\boldsymbol{g}}_t \rangle + \frac{M\eta_t^2\|\widehat{\boldsymbol{g}}_t\|^2}{2}$$

$$\leq \mathcal{L}(\boldsymbol{\theta}_t) - \eta_t(1 - \alpha)\|\boldsymbol{g}_t\|^2 + \frac{M\eta_t^2(1 + \alpha)^2\|\boldsymbol{g}_t\|^2}{2}$$

Rearranging the above equation, using the gradient norm bound in Equation (38) and the fact that $\alpha \in (0, 1)$, we get:

$$\left((1 - \alpha)\eta_t - \frac{M}{2}\eta_t^2\right)\|\nabla\mathcal{L}(\boldsymbol{\theta}_t)\|^2 \leq \mathcal{L}(\boldsymbol{\theta}_t) - \mathcal{L}(\boldsymbol{\theta}_{t+1}) + \left(\alpha M\eta_t^2 + \frac{\alpha^2}{2}M\eta_t^2\right)\|\boldsymbol{g}_t\|^2$$

$$< \mathcal{L}(\boldsymbol{\theta}_t) - \mathcal{L}(\boldsymbol{\theta}_{t+1}) + 6\alpha ML^2 e^{4\gamma K}\eta_t^2$$

where the condition $\eta_t < 2(1 - \alpha)/M$ is used to ensure $(1 - \alpha)\eta_t - M\eta_t^2/2 > 0$.

Summing up the above inequalities from $t = 1$ to $T$, we get:

$$\sum_{t=1}^{T}\left(\left((1 - \alpha)\eta_t - \frac{M}{2}\eta_t^2\right)\|\nabla\mathcal{L}(\boldsymbol{\theta}_t)\|^2\right) < \sum_{t=1}^{T}(\mathcal{L}(\boldsymbol{\theta}_t) - \mathcal{L}(\boldsymbol{\theta}_{t+1})) + 6\alpha ML^2 e^{4\gamma K}\sum_{t=1}^{T}\eta_t^2$$

$$= \mathcal{L}(\boldsymbol{\theta}_1) - \mathcal{L}(\boldsymbol{\theta}_T) + 6\alpha ML^2 e^{4\gamma K}\sum_{t=1}^{T}\eta_t^2$$

$$\leq \mathcal{L}(\boldsymbol{\theta}_1) - \mathcal{L}^* + 6\alpha ML^2 e^{4\gamma K}\sum_{t=1}^{T}\eta_t^2$$

where the last inequality is due to the fact that $\mathcal{L}^* \leq \mathcal{L}(\boldsymbol{\theta}_{T+1})$.

Dividing both sides by $\sum_{t=1}^{T}((1-\alpha)\eta_t - M\eta_t^2/2)$, we get:

$$\sum_{t=1}^{T}\left(\frac{2(1-\alpha)\eta_t - M\eta_t^2}{\sum_{t=1}^{T}(2(1-\alpha)\eta_t - M\eta_t^2)}\|\nabla\mathcal{L}(\boldsymbol{\theta}_t)\|^2\right) < \frac{2(\mathcal{L}(\boldsymbol{\theta}_1) - \mathcal{L}^*) + 12\alpha ML^2 e^{4\gamma K}\sum_{t=1}^{T}\eta_t^2}{\sum_{t=1}^{T}(2(1-\alpha)\eta_t - M\eta_t^2)}$$

By definition of $p_Z$ in Equation (35), which is used to select a final solution among $\{\boldsymbol{\theta}_1, \ldots, \boldsymbol{\theta}_T\}$, we know that:

$$\mathbb{E}_{p_Z}[\|\nabla\mathcal{L}(\boldsymbol{\theta}_Z)\|^2] = \sum_{t=1}^{T}(p_Z(t)\|\nabla\mathcal{L}(\boldsymbol{\theta}_t)\|^2)$$

$$= \sum_{t=1}^{T}\left(\frac{2(1-\alpha)\eta_t - M\eta_t^2}{\sum_{t=1}^{T}(2(1-\alpha)\eta_t - M\eta_t^2)}\|\nabla\mathcal{L}(\boldsymbol{\theta}_t)\|^2\right)$$

$$< \frac{2(\mathcal{L}(\boldsymbol{\theta}_1) - \mathcal{L}^*) + 12\alpha ML^2 e^{4\gamma K}\sum_{t=1}^{T}\eta_t^2}{\sum_{t=1}^{T}(2(1-\alpha)\eta_t - M\eta_t^2)}$$

$\square$

Now, let us consider a simple case where we use a constant step size, which gives us the following corollary:

**Corollary 2.** *Under the conditions in Theorem 5 except that we use constant step sizes:*

$$\eta_t = \min\left\{\frac{1-\alpha}{M}, \frac{1}{\sqrt{T}}\right\}, \quad t = 1, \ldots, T \tag{40}$$

*then with probability at least $1 - \delta$, we have:*

$$\mathbb{E}_{p_Z}[\|\nabla\mathcal{L}(\boldsymbol{\theta}_Z)\|^2] < \frac{2M(\mathcal{L}(\boldsymbol{\theta}_1) - \mathcal{L}^*)}{(1-\alpha)^2 T} + \frac{2(\mathcal{L}(\boldsymbol{\theta}_1) - \mathcal{L}^*) + 12\alpha ML^2 e^{4\gamma K}}{(1-\alpha)\sqrt{T}}$$

*Proof.* Since we are using a constant step size, by Theorem 5, we know that:

$$\mathbb{E}_{p_Z}[\|\nabla\mathcal{L}(\boldsymbol{\theta}_Z)\|^2] < \frac{2(\mathcal{L}(\boldsymbol{\theta}_1) - \mathcal{L}^*) + 12\alpha ML^2 e^{4\gamma K} T\eta_1^2}{T\eta_1(2(1-\alpha) - M\eta_1)}$$

By Equation (40), we have:

$$\frac{2(\mathcal{L}(\boldsymbol{\theta}_1) - \mathcal{L}^*) + 12\alpha ML^2 e^{4\gamma K} T\eta_1^2}{T\eta_1(2(1-\alpha) - M\eta_1)} \leq \frac{2(\mathcal{L}(\boldsymbol{\theta}_1) - \mathcal{L}^*) + 12\alpha ML^2 e^{4\gamma K} T\eta_1^2}{T(1-\alpha)\eta_1}$$

$$= \frac{2(\mathcal{L}(\boldsymbol{\theta}_1) - \mathcal{L}^*)}{T(1-\alpha)\eta_1} + \frac{12\alpha ML^2 e^{4\gamma K}\eta_1}{1-\alpha}$$

$$\leq \frac{2(\mathcal{L}(\boldsymbol{\theta}_1) - \mathcal{L}^*)}{T(1-\alpha)}\max\left\{\frac{M}{1-\alpha}, \sqrt{T}\right\} + \frac{12\alpha ML^2 e^{4\gamma K}}{(1-\alpha)\sqrt{T}}$$

$$< \frac{2M(\mathcal{L}(\boldsymbol{\theta}_1) - \mathcal{L}^*)}{(1-\alpha)^2 T} + \frac{2(\mathcal{L}(\boldsymbol{\theta}_1) - \mathcal{L}^*) + 12\alpha ML^2 e^{4\gamma K}}{(1-\alpha)\sqrt{T}}$$

$\square$

Note that an alternative result for constant step sizes can also be obtained from Theorem 6 in [8]:

**Theorem 6.** *Under Assumptions 1 and 2, for arbitrary constants $\alpha \in (0, 1)$ and $\delta \in (0, 1)$, suppose we use constant step sizes:*

$$\eta_t = \frac{\sqrt{2(\mathcal{L}(\boldsymbol{\theta}_1) - \mathcal{L}^*)}}{(1+\alpha)2Le^{2\gamma K}\sqrt{MT}} \tag{41}$$

and the sample size $N_t$ used for estimating $\widehat{\boldsymbol{g}}_t$ satisfies Equation (36), then with probability at least $1 - \delta$, we have:

$$\min_{t=1,\ldots,T} \|\nabla \mathcal{L}(\boldsymbol{\theta}_t)\|^2 \leq \frac{(1+\alpha)2Le^{2\gamma K}\sqrt{2M(\mathcal{L}(\boldsymbol{\theta}_1) - \mathcal{L}^*)}}{(1-\alpha)\sqrt{T}} \tag{42}$$

Although both Corollary 2 and Theorem 6 give a convergence rate of $O(1/\sqrt{T})$, the strategy in Theorem 6 requires extra computational effort to compute $\|\nabla L(\boldsymbol{\theta}_t)\|$ for $t = 1, \ldots, T$ in order to select the solution with the minimum gradient norm. Since $\|\nabla L(\boldsymbol{\theta})\|$ cannot be computed exactly, Monte Carlo estimation will incur additional approximation error. By contrast, the strategy in our analysis does not have such issues and Theorem 5 provides a general analysis on the convergence rate of the randomized SGD algorithm with consistent but biased gradient estimators, which allows using different step sizes.

For example, starting from Equation (37) and with the fact that:

$$\sum_{t=1}^{T} t = O(T^2), \quad \sum_{t=1}^{T} \sqrt{t} = O(T^{\frac{3}{2}}), \quad \sum_{t=1}^{T} t^{-\frac{1}{4}} = O(T^{\frac{3}{4}}), \quad \sum_{t=1}^{T} t^{-\frac{1}{2}} = O(T^{\frac{1}{2}})$$

one can easily verify that using the following increasing step sizes:

$$\eta_t = \min\left\{\frac{1-\alpha}{M}, \frac{\sqrt{t}}{T}\right\}, \quad t = 1, \ldots, T$$

or decreasing step sizes:

$$\eta_t = \min\left\{\frac{1-\alpha}{M}, \frac{1}{(tT)^{1/4}}\right\}, \quad t = 1, \ldots, T$$

will give us a similar convergence rate of $O(1/\sqrt{T})$.

Finally, if we would like to make a stronger assumption that the loss function $\mathcal{L}(\boldsymbol{\theta})$ is strongly convex, then we can obtain a stronger result that $\boldsymbol{\theta}_T$ converges to the optimal solution $\boldsymbol{\theta}^*$ in $L_2$-norm with a convergence rate of $O(1/T)$.

**Assumption 3.** *The loss function $\mathcal{L}(\boldsymbol{\theta})$ is $J$-strongly convex (with $J > 0$):*

$$\forall \boldsymbol{\theta}_1, \boldsymbol{\theta}_2 \in \Theta, \mathcal{L}(\boldsymbol{\theta}_2) - \mathcal{L}(\boldsymbol{\theta}_1) \geq \langle \nabla \mathcal{L}(\boldsymbol{\theta}_1), \boldsymbol{\theta}_2 - \boldsymbol{\theta}_1 \rangle + \frac{J}{2}\|\boldsymbol{\theta}_2 - \boldsymbol{\theta}_1\|^2.$$

*and the unique optimum of $\mathcal{L}(\boldsymbol{\theta})$ is $\boldsymbol{\theta}^*$.*

**Theorem 7.** *Under Assumptions 1 and 3, for arbitrary constant $\delta \in (0,1)$, suppose that $J \leq 2Le^{2\gamma K}/\|\boldsymbol{\theta}_1 - \boldsymbol{\theta}^*\|$ and we use decreasing step sizes:*

$$\eta_t = \frac{1}{(J - J/(2T))t}, \quad t = 1, \ldots, T \tag{43}$$

*and the sample size $N_t$ used for estimating $\widehat{\boldsymbol{g}}_t$ satisfies:*

$$N_t \geq \frac{128L^2T^2e^{8\gamma K}(1 + 4\log(2T/\delta))}{J^2\|\boldsymbol{g}_t\|^2} \tag{44}$$

*then with probability at least $1 - \delta$, we have:*

$$\|\boldsymbol{\theta}_T - \boldsymbol{\theta}^*\| \leq \frac{4L^2e^{4\gamma K}}{T}\left[\frac{(2 + J/T)^2 + J^2(2 - 1/T)}{J^2(2 - 1/T)^2}\right] \tag{45}$$

Intuitively, Theorem 7 implies that when the loss function $\mathcal{L}(\boldsymbol{\theta})$ is strongly convex in $\boldsymbol{\theta}$ with $\boldsymbol{\theta}^*$ being the optimal solution, then under some conditions on the sample sizes for estimating the gradients and step sizes for updating the parameters, the output of the SGD algorithm $\boldsymbol{\theta}_T$ will converge to $\boldsymbol{\theta}^*$ with a convergence rate of $O(1/T)$.

When the loss function $\mathcal{L}(\boldsymbol{\theta})$ is convex but not strongly convex, we have the following theorem showing a typical convergence rate of $O(1/\sqrt{T})$:

**Assumption 4.** *The loss function $\mathcal{L}(\boldsymbol{\theta})$ is convex and the parameter space has finite diameter $D$:* $\sup_{\boldsymbol{\theta}_1,\boldsymbol{\theta}_2 \in \Theta} \|\boldsymbol{\theta}_1\| = D$. *Let $\boldsymbol{\theta}^* \in \arg\min_{\boldsymbol{\theta} \in \Theta} \mathcal{L}(\boldsymbol{\theta})$.*

**Theorem 8.** *Under Assumptions 1 and 4, for arbitrary constant $\delta \in (0,1)$, suppose we use decreasing step sizes $\eta_t = 1/\sqrt{t}$ for $t = 1, \ldots, T$ and the sample size $N_t$ used for estimating $\widehat{\boldsymbol{g}}_t$ satisfies:*

$$N_t \geq \frac{32L^2 T e^{8\gamma K}(1 + 4\log(2T/\delta))}{\|\boldsymbol{g}_t\|^2} \tag{46}$$

*then with probability at least $1 - \delta$, we have:*

$$f(\overline{\boldsymbol{\theta}}_T) - f(\boldsymbol{\theta}^*) \leq \frac{1}{\sqrt{T}} \left[ D^2 + 2L^2 e^{4\gamma K} \left( 1 + \left( 1 + \frac{1}{\sqrt{T}} \right)^2 \sqrt{1 + \frac{1}{T}} \right) \right]$$

*where $\overline{\boldsymbol{\theta}}_T := \frac{1}{T} \sum_{t=1}^T \boldsymbol{\theta}_t$.*

The above theorems follow from the sample complexity bound in Theorem 4 and the results in [8] (Theorem 2 with constant $\rho = J/2$ and Theorem 5 with constants $\rho = 1, c = 1$), which we refer to for a detailed proof. Note that the condition on $J$ in Theorem 7 is optional and without the condition, we can obtain the same convergence rate at the cost of a cumbersome form in the R.H.S. of Equation (45).

# D  Additional Experimental Details

## D.1  2-D Synthetic Data Experiments

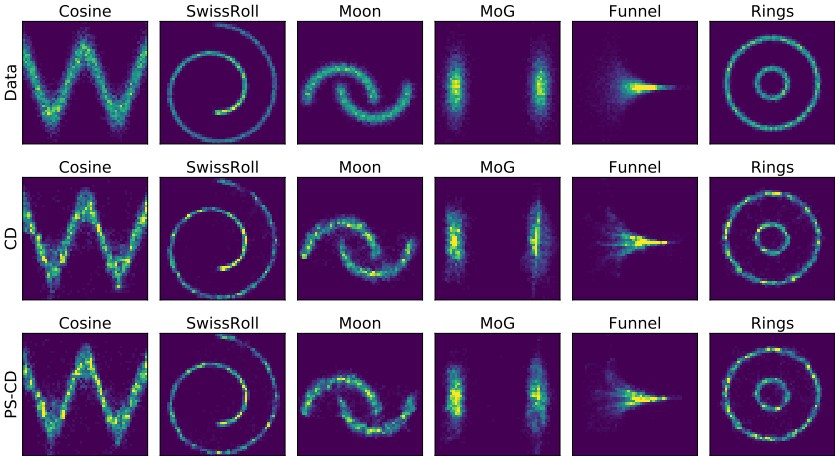

Figure 2: Histograms of samples from the data distribution (top), CD (middle) and PS-CD (bottom).

Table 3: Maximum mean discrepancy (MMD, multiplied by $10^4$) results on six 2-D synthetic datasets. Lower is better. CD denotes contrastive divergence algorithm, and PS-CD denotes the pseudo-spherical contrastive divergence algorithm (with $\gamma = 1.0$).

| Method | Cosine | Swiss Roll | Moon | MoG | Funnel | Rings |
|--------|--------|-----------|------|-----|--------|-------|
| CD | $1.20 \pm 0.45$ | $3.39 \pm 0.48$ | $0.64 \pm 0.11$ | $3.01 \pm 0.58$ | $\mathbf{1.56} \pm 0.65$ | $2.79 \pm 0.63$ |
| PS-CD | $\mathbf{0.86} \pm 0.12$ | $\mathbf{0.89} \pm 0.39$ | $\mathbf{0.12} \pm 0.04$ | $\mathbf{1.78} \pm 0.35$ | $2.34 \pm 0.45$ | $\mathbf{2.02} \pm 0.32$ |

## D.2  Understanding the Effects of Different $\gamma$ Values in 1-D Examples

In this section, we aim to provide insights on the effects of different $\gamma$ values with 1-D toy experiments. Specifically, we use an EBM with a quadratic energy function (corresponding to a Gaussian distribution):

$$E_{\mu,\sigma}(x) = \frac{(x - \mu)^2}{2\sigma^2}, \ q_{\mu,\sigma}(x) \propto \exp(-E_{\mu,\sigma}(x)) \tag{47}$$

where $\mu$ and $\sigma$ are two trainable parameters.

First, we show that when the real data distribution is also a Gaussian distribution such that the model is well-specified, then different $\gamma$ values will induce the same optimal distribution since they are *strictly proper*. To verify this property, we visualize the objective landscape in Figure 3.

Second, we use the same quadratic energy function to fit a mixture of Gaussians. We visualize the objective landscapes in Figure 4, which shows that when the model is mis-specified, different objectives will exhibit different modeling preferences (inducing different solutions). This corresponds to the practical scenarios, where such property enables us to flexibly specify different inductive biases to make tradeoff among various modeling factors such as diversity/quality.

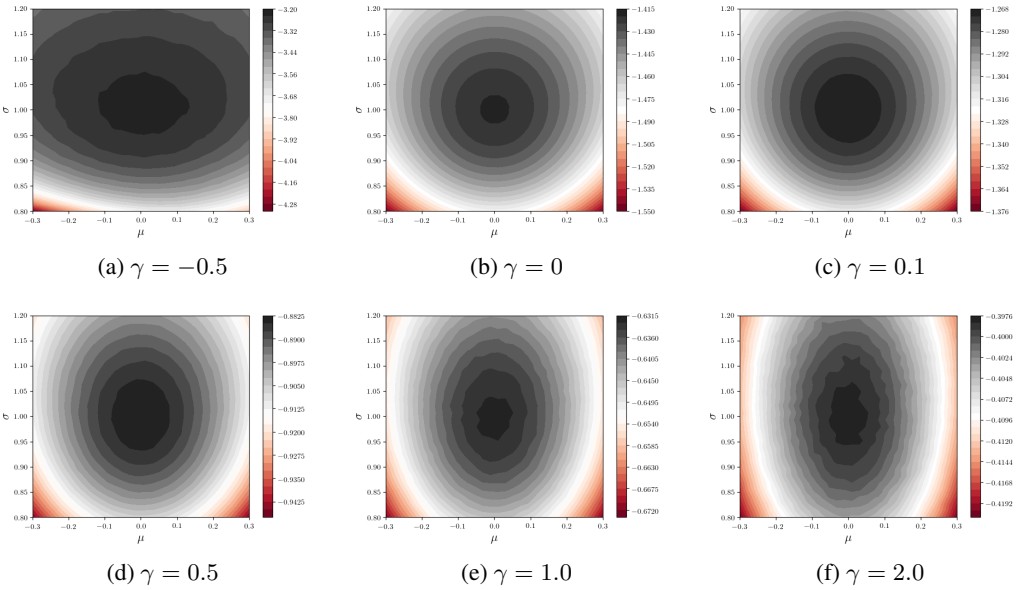

Figure 3: Visualization of different objective landscapes for model well-specified scenarios. $\gamma = 0$ corresponds to the logarithm scoring rule (MLE) and other values correspond to the $\gamma$-scoring rules.

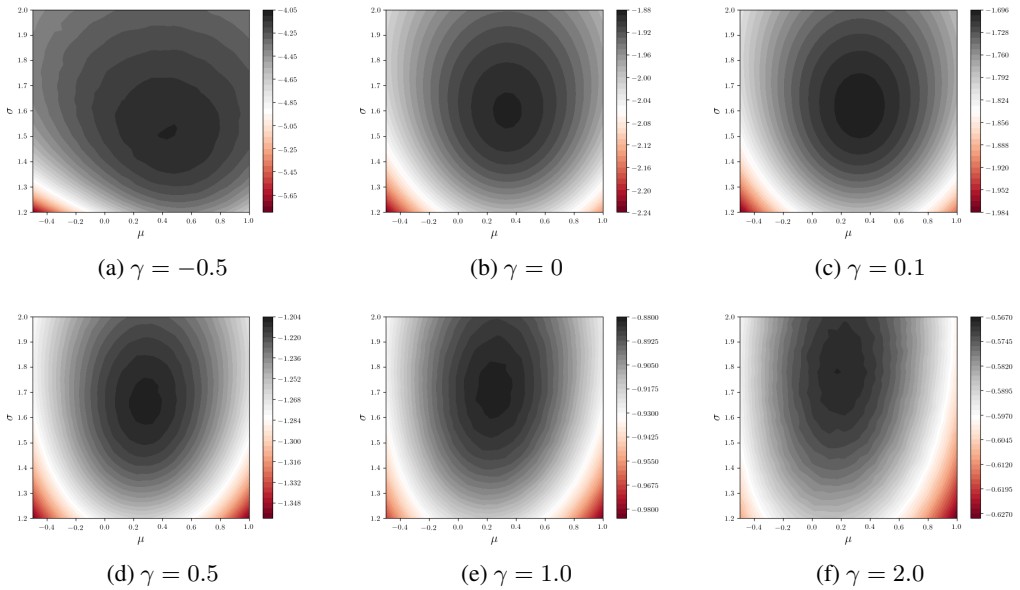

Figure 4: Visualization of different objective landscapes for model mis-specified scenarios. $\gamma = 0$ corresponds to the logarithm scoring rule (MLE) and other values correspond to the $\gamma$-scoring rules.

## D.3 Understanding the Effects of Different $\gamma$ Values in Image Generation

Although FID has been the most popular evaluation metric for image generative models, it is problematic since it summarizes the difference between two distributions into a single number and fails to separate important aspects such as fidelity and diversity [61]. To better demonstrate the modeling flexibility brought by the proposed PS-CD framework, we conduct experiments on CIFAR-10 dataset using a set of more indicative and reliable metrics proposed by [61] to evaluate the effects of $\gamma$ from various perspectives.

Table 4: Effects of $\gamma$ on CIFAR-10 image generation. We use the same image embeddings (activations of a pre-trained inception network) to compute these metrics and FID to ensure consistency. We briefly introduce these metrics here and refer to [61] for accurate descriptions and mathematical definitions. Denote data distribution as $P(X)$ and model distribution as $Q(X)$. Based on manifold estimation, *Precision* is defined as the portion of $Q(X)$ that can be generated by $P(X)$ and *Recall* is symmetrically defined as the portion of $P(X)$ that can be generated by $Q(X)$; *Density* improves upon *Precision* to count how many real-sample neighbourhood spheres contain a certain fake sample; *Coverage* improves upon *Recall* to measure the fraction of real samples whose neighbourhoods contain at least one fake sample.

|  | Density | Coverage | Precision | Recall | FID |
|---|---|---|---|---|---|
| CD ($\gamma = 0$) | 0.693 | 0.601 | 0.798 | 0.368 | 37.90 |
| PS-CD ($\gamma = -0.5$) | 0.906 | 0.691 | 0.848 | 0.360 | 27.95 |
| PS-CD ($\gamma = 0.5$) | 0.772 | 0.634 | 0.819 | 0.352 | 35.02 |
| PS-CD ($\gamma = 1.0$) | 0.929 | 0.694 | 0.853 | 0.341 | 29.78 |
| PS-CD ($\gamma = 2.0$) | 0.932 | 0.652 | 0.861 | 0.351 | 33.19 |

From Table 4, we have some interesting observations: (1) PS-CD with $\gamma = -0.5$ and $\gamma = 1.0$ get best FID scores because they can simultaneously achieve good balance among these metrics (e.g., high *Density* and *Coverage* score); (2) By contrast, PS-CD with $\gamma = 2.0$ achieves the highest *Density* score but a relatively low *Coverage* score, which potentially leads to a slightly worse FID; (3) Many members in the PS-CD family showed superior performance over traditional contrastive divergence in most metrics, demonstrating the potential of our method. Just like various $f$-divergences used in generative modeling, different members in the PS-CD family can represent complicated inductive bias in practice (although being strictly proper in model well-specified case). Since these single-valued evaluation metrics measure the generative performance in a complicated way, we think it is normal that the change of $\gamma$ is not monotone to the change of each metric. For specific application scenarios, we may mainly care about a certain metric and we should choose $\gamma$ accordingly.

We would like to emphasize that, a major contribution of our paper is opening the door to a new family of EBM training objectives and enabling us to flexibly specify modeling preferences, without introducing additional computational cost compared to CD (unlike adversarial training in $f$-EBM).

### D.4 Image Generation Samples for PS-CD

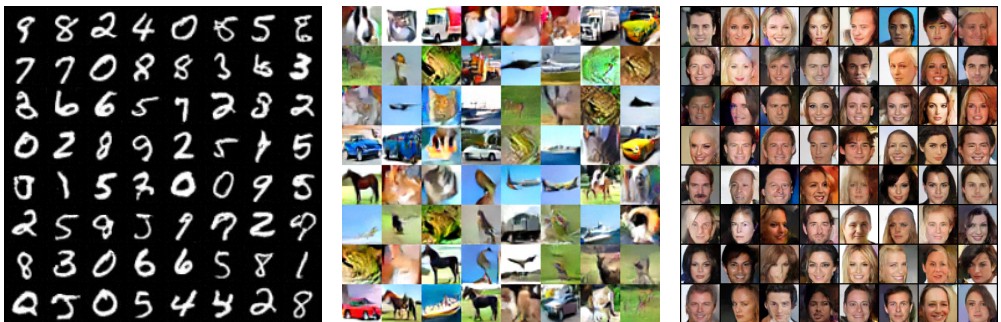

Figure 5: MNIST, CIFAR-10 and CelebA samples for PS-CD ($\gamma = 1.0$).

## D.5 Training Details

To keep a fair comparison, all the compared methods use the same architecture to implement the energy function, except that $f$-EBMs require an additional variational function that uses the same architecture as the energy function. The architectures used for CIFAR-10 ($32 \times 32$) and CelebA ($64 \times 64$) datasets are shown in Figure 6. We use leaky-ReLU non-linearity with default leaky factor 0.2 throughout the architectures (between all the convolution layers). Following [17, 89], we apply spectral normalization/$L_2$ regularization (on the outputs of the models) with coefficient 1.0 to improve the stability.

For CIFAR-10, to keep a fair comparison, we use the same sampling strategy as [17, 89], where a sample replay buffer is employed to improve the mixing of Langevin dynamics. Specifically, we use 60 steps Langevin dynamics together with a sample replay buffer of size 10000 to produce samples in the training phase. In each Langevin step, we use a step size of 10.0 and a random noise with standard deviation of 0.005.

For CelebA, which has a higher data dimension, we use the sampling strategy in [66] to improve the efficiency of sampling, where we always start the Markov chains from a fixed uniform distribution and run a fixed number of Langevin steps (100) with a constant step size.

For all the experiments, we use Adam optimizer to optimize the parameters of the energy function. In each training iteration, we use a batch size of $128$ for CIFAR-10 and $64$ for CelebA. We run the PS-CD algorithms for about 50K iterations of parameter updates for CIFAR-10 and about 100K iterations for CelebA.

For computational cost, the CIFAR-10 experiments take about 48 hours on 4 Titan Xp GPUs, while the CelebA experiments take about 16 hours since we learn non-convergent short-run MCMC.

| 3x3 Conv2d, 128 |
| ResBlock Down 128 |
| ResBlock 128 |
| ResBlock Down 256 |
| ResBlock 256 |
| ResBlock Down 256 |
| ResBlock 256 |
| Global Sum Pooling |
| Dense $\rightarrow$ 1 |

(a) CIFAR-10 ($32 \times 32$)

| 3x3 Conv2d, 64 |
| 4x4 Conv2d, 128 |
| 4x4 Conv2d, 256 |
| 4x4 Conv2d, 512 |
| 4x4 Conv2d, 512 |
| 4x4 Conv2d, 1 |

(b) CelebA ($64 \times 64$)

Figure 6: Network architectures.

## D.6 OOD Detection & Robustness to Data Contamination

Table 5: OOD Detection results (AUROC score) for models trained on CIFAR-10.

| OOD Dataset | PixelCNN++ | Glow | CD | PS-CD |
|---|---|---|---|---|
| SVHN | 0.32 | 0.24 | 0.43 | 0.56 |
| Textures | 0.33 | 0.27 | 0.36 | 0.44 |
| Uniform/Gaussian | 1 | 1 | 1 | 1 |
| CIFAR-10 Interpolation | 0.71 | 0.59 | 0.63 | 0.68 |
| CelebA | / | / | 0.51 | 0.58 |

Table 6: Training EBMs under data contamination on CIFAR-10. We measure the change of FID score after training with the contaminated dataset.

| | Pretrained Model | CD 1000 Steps | CD 2000 Steps | PS-CD 1000 Steps | PS-CD 2000 Steps |
|---|---|---|---|---|---|
| FID | 68.77 | 95.56 | 300.89 | 59.78 | 57.24 |

To show the practical advantage of PS-CD in face of data contamination, we further conduct experiments on MNIST and CIFAR-10 datasets, where we use random uniform noise as the contamination distribution and the contamination ratio is 0.1 (i.e. 10% images in the training set are replaced with

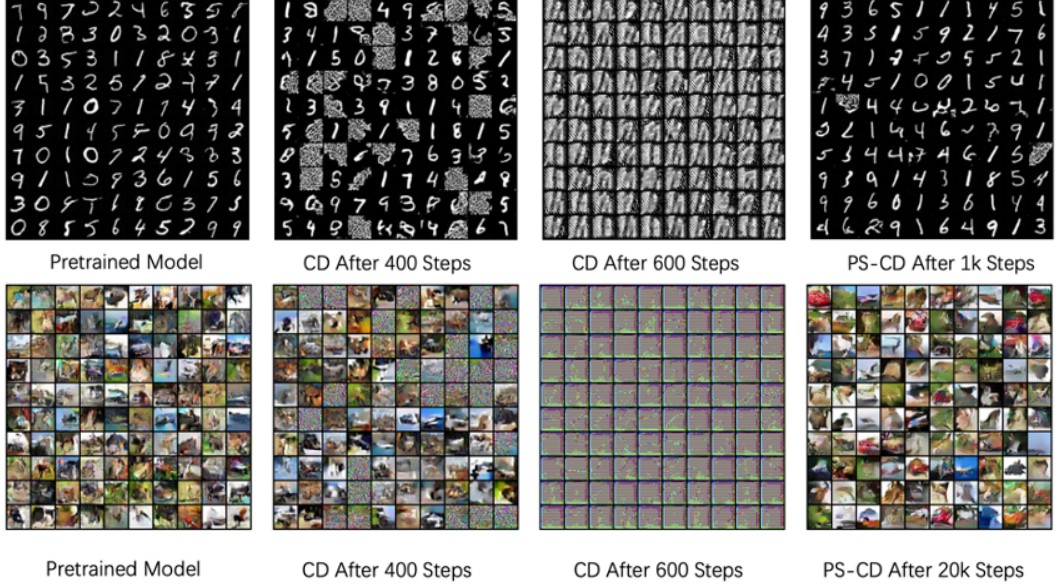

| Pretrained Model | CD After 400 Steps | CD After 600 Steps | PS-CD After 1k Steps |
|---|---|---|---|

| Pretrained Model | CD After 400 Steps | CD After 600 Steps | PS-CD After 20k Steps |
|---|---|---|---|

Figure 7: Samples after training with the contaminated dataset on MNIST and CIFAR-10.

random noise). After a warm-up pretraining (when the model has some OOD detection ability), we train the model with the contaminated data and measure the training progress of CD and PS-CD.

As shown in Figure 7, CD gradually generates more and more random noise and diverge after a few training steps, while PS-CD is very robust. In particular, as shown in Table 6, for a slightly pretrained unconditional CIFAR-10 model (a simple 5-layer CNN with FID of 68.77), we observe that the performance of CD degrades drastically in terms of FID, while PS-CD can continuously improve the model even using the contaminated data.

We believe that robustness to data contamination is a valuable property for modern deep generative models and there is actually a natural interpretation for the robustness of PS-CD. Compared to CD, there is an extra weight term before the gradient of the energy: $\frac{\exp(-\gamma E_{\boldsymbol{\theta}}(\boldsymbol{x}_i))}{\sum_j \exp(-\gamma E_{\boldsymbol{\theta}}(\boldsymbol{x}_j))} \nabla_{\boldsymbol{\theta}} E_{\boldsymbol{\theta}}(\boldsymbol{x}_i)$ (the first term in Eq. (19)). Suppose $\boldsymbol{x}_i \sim \omega$ is a noise data from the contaminated distribution $\tilde{p}$ in a batch of samples, for a model with OOD detection ability, it will assign a much higher energy to $\boldsymbol{x}_i$ than normal data and the weight before $\nabla_{\boldsymbol{\theta}} E_{\boldsymbol{\theta}}(\boldsymbol{x}_i)$ will be close to zero. In short, PS-CD naturally integrates the OOD detection ability of EBMs into the training process, which then leads to robustness to data contamination.