# OpenReview forum: "Pseudo-Spherical Contrastive Divergence"
_NeurIPS.cc/2021/Conference — NeurIPS 2021 Poster_

### Official Review · Reviewer_4NoL · 2021-07-14

**Rating:** 6
**Confidence:** 4

**Summary:**

The paper introduces a class of divergence to learn EBMs. The class of divergence includes KL as a special case. The paper approximates the gradient of the class of divergence by self-normalized importance sampling and analyzes the convergence when optimizing with the approximate gradient. Experiments are conducted to evaluate the influence of $\gamma$.

**Limitations And Societal Impact:**

Yes

**Main Review:**

Pros:
The idea is relatively novel, which extends CD to PS-CD. Under some $\gamma$ PS-CD outperforms CD. The paper is clearly written.

Questions:
1. Do you have a sense how $\gamma$ influences the FID results in Table 1? How to choose a best $\gamma$ in practice?

2. In practice, the training of CD is unstable, e.g., the training is likely to diverge (see [a], Appendix H.3). Will the PS-CD overcome these issues?


[a] YOUR CLASSIFIER IS SECRETLY AN ENERGY BASED MODEL AND YOU SHOULD TREAT IT LIKE ONE

**Time Spent Reviewing:**

4

---

> ### Author Response · Authors · 2021-08-10
> **Author Response**
>
> Thanks for your valuable feedback! Besides the general response with new experimental results and discussions, we have addressed your questions below:
>
> 1) We think how to choose the best $\gamma$ or select the most suitable proper scoring rule in the family of PS score for a specific application or evaluation metric (like FID) is still an open question. Empirically we found the classic spherical scoring dule ($\gamma=1$) and our negative extension works well. In practice, we can conduct experiments on synthetic data or subsets of large datasets to obtain a set of candidates quickly and then test them on full datasets.
>
> 2) Yes, we observe that PS-CD shows better stability and robustness, especially when dealing with data contamination. Please look at the general response for new experimental results and discussions on OOD detection and robust EBM learning under data contamination.

---

> > ### Comment · Reviewer_4NoL · 2021-08-26
> > **Thanks for your reply**
> >
> > I decide to keep my score. Meanwhile, Q2 is not fully addressed and I hope Q2 could be discussed more in the final version (whether the training will diverge with sufficient long iterations?)

---

> > > ### Author Response · Authors · 2021-08-26
> > > **More discussions on Q2**
> > >
> > > Thanks for the feedback! We would like to provide more discussion on Q2 and we will also include it in the revised version:
> > >
> > > From Appendix H.3 of [a], we noticed that the instability of traditional Persistent CD is because: occasionally throughout training a sample will be drawn from the replay buffer that has a considerably higher-than average energy, which then causes the gradients w.r.t this example much larger than normal and the model will diverge. Similar to our new experiments and interpretation on "robustness to data contamination", in principle, our PS-CD framework will also be more robust to such situation.
> > >
> > > Specifically, for samples from the model distribution (replay buffer when using long-run MCMC), compared to CD, there is also an extra weight term before the gradient of the energy (the **second** term in Eq. (19)):
> > >
> > > $$
> > > \frac{\exp(- \gamma E_\theta(x_i^-))}{\sum_j \exp(- \gamma E_\theta(x_j^-))} \nabla_\theta E_\theta(x_i^-), \quad x_1^-, \ldots, x_n^- \sim q_\theta
> > > $$
> > >
> > > When the same situation happens (i.e. one of the model samples $x_i^-$ from the replay buffer has a much higher-than average energy), the corresponding weight before the energy gradient $\nabla_\theta E_\theta(x_i^-)$ will be close to zero and hence the negative effect of such an outlier will be removed. By contrast, in traditional CD (corresponding to $\gamma=0$), the weight term before the energy gradient is always $1/n$ (constant) and thus any outlier has a big impact on the optimization, while PS-CD is naturally robust to data contamination/outliers in both real data and model samples.
> > >
> > > Thanks for the suggestion and we hope this can fully address your concern. We will provide more detailed discussion on this point in the revised version.
> > >
> > > [a] YOUR CLASSIFIER IS SECRETLY AN ENERGY BASED MODEL AND YOU SHOULD TREAT IT LIKE ONE

---

### Official Review · Reviewer_y75x · 2021-07-16

**Rating:** 7
**Confidence:** 3

**Summary:**

The authors propose a spectrum of objectives for learning energy-based models that generalise maximum likelihood. These objectives--known as pseudo-spherical scoring rules--are similar to (persistent) contrastive divergence in terms of their implementation and computational cost, but have an extra degree of flexibility that can boost downstream performance for e.g. image generation.

**Limitations And Societal Impact:**

I am happy with the author's discussion of limitations and societal impact.

**Main Review:**

**Overall verdict**

This paper contains some solid results, with no obvious technical shortcomings. Personally, I didn't find the method especially exciting and the experiments were slightly limited, leaving me with concerns about the significance/impact of the paper. Nevertheless, I lean towards acceptance.

**Originality + Quality + Significance**

This proposed method, PS-CD, is similar in spirit to f-EBM [1] insofar as it offers a spectrum of losses that are suitable for training energy-based models. The key differences to f-EBM are:

1) The losses are not based on f-divergences, but rather spherical scoring rules, which correspond to Bregman divergences.
2) Unlike f-EBM, PS-CD does not require us to solve a minimax problem, and therefore has the same computational cost as persistent contrastive divergence.

I think these contributions are solid, and the accompanying theory & experiments are sufficient to convince me that the method is correct and potentially useful.

I don't think this paper constitutes any kind of fundamental advance for energy-based modelling, but it provides an extra 'knob to twist' when learning EBMs, which *may* give us more control over the kinds of statistical patterns extracted by the EBM. This final claim is supported by the experiments, where we see modest improvements in FID scores compared to f-EBM.

A couple of issues/questions (in order of importance):

1) I found the experiments to be quite limited (2d synthetic density estimation +  image generation on 3 datasets). I would have been much more convinced if you had considered other applications e.g. out-of-distribution detection or joint energy-based modelling and classification as in [5].

2) In line 81, you cite [2] as a 'recent advance...to obtain samples from an EBM'. This is misleading, since the point of that paper was that short-run "samples" are not actually valid samples from the EBM (although they can still be high quality). I think this point needs emphasising in the paper, and you should make it clear when reporting results whether or not the samples are likely to be valid. I also think its good practice to report the quality of samples that are obtained after running the MCMC sampler for a long time (e.g. tens of thousands of time steps).

3) I think you should clarify in the main text exactly how your convergence analysis builds on [3] & [4], which you describe as 'prior theoretical works for analysing SGD'. Is it fair to describe theorem 5 as essentially an application of the existing theory? Or are there techniques used in your proofs that are sufficiently non-obvious that readers will learn something by reading them?

4) The theoretical results rely on the sample size growing exponentially with K (which bounds the energy). I imagine that actually using this number of samples is not possible in practice since K would be too large. Do the authors agree? What range of energy values are observed in the experiments? If I am right, then this means the theoretical guarantees have quite limited practical relevance.

5) Equation 19 relies on self-normalised importance sampling. Have you tried measuring the variance / effective-sample-size of this estimator?

**Clarity + prior work**

I think the paper is very well structured & written, and that the references to prior work are extremely thorough. There has been a lot of literature published on EBMs in the last couple of years, and I think the authors cite most of the key papers.

** Minor points **

- Personally, I don't think Theorems 2 & 3 warrant the title 'Theorem'. Just stating the equations would be sufficient for me.

** References **

[1] Lantao Yu, Yang Song, Jiaming Song, and Stefano Ermon. Training deep energy-based models with f-divergence minimization.  InInternational Conference on Machine Learning, pages10957–10967. PMLR, 2020.

[2] Erik Nijkamp, Mitch Hill, Song-Chun Zhu, and Ying Nian Wu.  Learning non-convergent non-persistent short-run mcmc toward energy-based model. InAdvances in Neural Information Processing Systems, pages 5233–5243, 2019

[3] Jie Chen and Ronny Luss.  Stochastic gradient descent with biased but consistent gradient estimators.arXiv preprint arXiv:1807.11880, 2018

[4] Saeed Ghadimi and Guanghui Lan. Stochastic first-and zeroth-order methods for nonconvex stochastic programming.SIAM Journal on Optimization, 23(4):2341–2368, 2013.

[5] Grathwohl, Will, et al. "Your classifier is secretly an energy based model and you should treat it like one." arXiv preprint arXiv:1912.03263 (2019).

**Time Spent Reviewing:**

5

---

> ### Author Response · Authors · 2021-08-10
> **Author Response**
>
> Thanks for your valuable feedback! Besides the general response with new experimental results and discussions, we have addressed your questions below:
>
> 1) Other applications such as OOD detection: Thanks for the suggestion! We have conducted new experiments on OOD detection and robust EBM learning under data contamination, which can be found in the general response.
>
> 2) We will revise our paper accordingly when introducing [1] to emphasize that short-run "samples" are not actually valid samples from the EBM (although they can still be high quality). In the meantime, as a general EBM learning framework, we tested our method in conjunction with both short-run and long-run MCMC. Specifically, in the training and evaluation process, we use the short-run MCMC strategy [1] for CelebA and long-run MCMC replay buffer strategy [2] for CIFAR-10. We will update our paper to emphasize this when reporting the results.
>
> 3) We will clarify in the main text how the convergence analysis builds on [3] and [4]. Basically, after obtaining the sample complexity bound using vector Bernstein inequality (which the convergence analysis also builds on), we apply similar proving strategies in [3] and [4] in our context to analyze the convergence of PS-CD.
>
> 4) In practice, we typically use L2 regularization (also used in [2] and [5]) on the output of the energy function to improve stability. Thus the value K that upper bounds the energy value is usually very small (less than 1 in our experiments). Thus we think our theoretical results are still meaningful.
>
> 5) Measuring the variance / effective-sample-size of the estimator: We did not investigate this property in our experiments. Empirically we found that using a common batch size as in contrastive divergence (no self-normalized IS) such as 64 or 100 works well in practice. We will incorporate corresponding discussions in the updated version.
>
> [1] Learning non-convergent non-persistent short-run mcmc toward energy-based model.
>
> [2] Implicit Generation and Modeling with Energy-Based Models
>
> [3] Stochastic gradient descent with biased but consistent gradient estimators.
>
> [4] Stochastic first-and zeroth-order methods for nonconvex stochastic programming
>
> [5] Training deep energy-based models with f-divergence minimization

---

> > ### Comment · Reviewer_y75x · 2021-08-22
> > **Increased score**
> >
> > Thank you for the thorough response! I feel that all of my concerns have been addressed.
> >
> > The additional experiments have pushed me from 'weak accept' to 'accept'. In particular, I think the 'robustness to data contamination' results are a strong selling-point for the paper.

---

> > > ### Author Response · Authors · 2021-09-02
> > > **Thanks a lot for your support!**
> > >
> > > We are glad to hear that all of your concerns have been addressed by our response!
> > >
> > > We really appreciate your support and we will incorporate your valuable suggestions and the new experimental results/discussions in the revised version.

---

### Official Review · Reviewer_mWGE · 2021-07-16

**Rating:** 6
**Confidence:** 4

**Summary:**

This paper presents a generalization of the Contrastive Divergence that uses pseudo-spherical scoring rules to define a new family of loss functions for generative modeling. This family of objective functions has an auxiliary loss parameter that can balance the trade-off between diversity and quality. The method is applied to learn EBMs on image benchmark datasets.

**Limitations And Societal Impact:**

The authors have adequately addressed the limitations and societal impact of their work.

**Main Review:**

The main contribution of this paper is to define a new framework for EBM learning by situating the classical CD objective function within a continuous spectrum of loss functions. While a similar concept has been used to generalize EBM learning using $f$-divergences, the new framework presented in the present work is distinct from prior formulations. Furthermore, the PS-CD method proposed in this work encompasses CD as a special case for a certain hyperparameter setting. The theoretical derivation of a novel EBM loss is quite original and the strongest aspect of the paper.

The proposed method is evaluated on benchmark datasets commonly used in generative modeling. The authors show that adjusting the value of $\gamma$ to be non-zero tends to give significantly better results than standard CD learning corresponding to $\gamma = 0$. The authors observe good results for both positive and negative non-zero values of $\gamma$, and I am unsure of the explanation behind this. I am not sure that conditional Cifar-10 is the best benchmark because it is much less common than unconditional Cifar-10. Unconditional Cifar-10 might be more appropriate since there are a wider variety of existing benchmarks for comparison. In the experiments presented, the PS-CD learning method appears to yield some moderate improvement in the quality of generated images. I am curious if the authors have techniques for tuning $\gamma$ beyond trying out a few different options.

**Overall**: The proposed PS-CD learning method is original and interesting. Although the empirical results might not show dramatic practical benefits of the learned model, the work is nonetheless a novel contribution and the generalizations of CD used in this work might influence future practitioners.

**Time Spent Reviewing:**

5 hours

---

> ### Author Response · Authors · 2021-08-10
> **Author Response**
>
> Thanks for your valuable feedback! Besides the general response with new experimental results and discussions, we have addressed your comments below:
>
> As an initial investigation to pseudo-spherical scoring rule with negative $\gamma$, we surprisingly found its effectiveness in our empirical study. We then established its connection to Renyi entropy regularized maximum likelihood in Lemma 2. Just like the effects of different $f$-divergence, we agree that theoretical analysis on the inductive bias of different PS scores for generative modeling deserves further study.
>
> We use CIFAR-10 conditional generation as one of the experiments in order to have a direct comparison with $f$-EBM [1], which also provides a spectrum of proper objectives and generalizes CD. We believe that such comparison is most meaningful as all methods (CD, PS-CD, f-EBM) strictly share the same network architecture and hyper-parameters (e.g. in Langevin dynamics sampling). We also conducted unconditional EBM learning on CelebA dataset to demonstrate the effectiveness in both scenarios (conditional and unconditional energy-based modeling).
>
> Strategy for tuning $\gamma$ beyond trying out a few different options: Currently, we tune the parameter $\gamma$ as a hyper-parameter and a common strategy for quickly obtaining a suitable $\gamma$ is conducting experiments on synthetic data or subsets of large datasets. Nevertheless, we believe that further theoretical study on the effects of $\gamma$ (e.g. on robustness to data contamination in existing literatures) will shed light on how to choose $\gamma$ properly.
>
> In the general response, we also provided new experimental results on OOD detection and robust EBM learning under data contamination, which we believe will better demonstrate the practical benefits of our framework.
>
> [1] Training deep energy-based models with f-divergence minimization.

---

### Official Review · Reviewer_CEzj · 2021-07-24

**Rating:** 7
**Confidence:** 3

**Summary:**

This paper provides an efficient framework to estimate energy based model (EBM).
While conventional KL or general f-divergence-based contrastive divergence learning suffers from instability and non-convergence issues, the proposed framework namely pseudo-spherical contrastive divergence (PS-CD) can be implemented as SGD type algorithm whose sample complexity and convergence property are provided.
PS-CD includes the conventional CD as a special case and thus provides gradient-based optimization framework for CD.
Experiments to evaluate the effectiveness of the proposed approach are conducted with both synthetic and image benchmark datasets.

**Limitations And Societal Impact:**

While it provides a general framework for learning EBMs, it is not as good as SOTA methods in terms of FID score (although I believe that it is not appropriate to evaluate generative models with a single score).
I believe that there is no negative social impact from this research.


**Main Review:**

Originality: The proposed method introduces the PS score in statistical decision theory for learning EBM. The proposed method, PS-CD, is defined as 1/\gamma times the negative logarithmic PS scoring rule for parameter \gamma. The advantage of PS-CD is that a consistent estimator of its gradient can be easily computed, allowing EBMs to be trained by gradient methods. This is an advantage of using the PS scoring rule, an approach that has not been seen in previous learning methods.
Since PS-CD reproduces ordinary CD by adjusting the parameter \gamma, it can be interpreted that the proposed method is a generalization of CD and also provides learning by gradient method.

Quality: This paper technically sounds. Although the main contribution of the paper is the proposal of a learning framework and the investigation of its theoretical properties, minimal experimental results and discussion are also presented to evaluate the effectiveness proposed method.

Clarity: This paper is well written. It clearly explains why a more general framework is needed (what are issues with conventional methods), the process of deriving the proposed method, and its theoretical considerations.

Significance:For the learning of EBMs discussed in this paper, the CD method, which is the maximum likelihood learning based on KL divergence, and the f-EBM, which minimizes the more general f-divergence, have been usually used. These methods had difficulties such as unstable learning and complex variational computation. The presented work that avoids these issues and enable learning by the gradient method is considered to have a high impact (
the derived sample complexity and convergence rate show that the proposed method can theoretically achieve the similar performance as that of learning by ordinary SGD).

Questions:
- In the experiment, the authors discuss the effect of the parameter \gamma in the case of model misspecification. How much of the effect of the model-misspecification can be absorbed by adjusting the \gamma? From the experimental results, I think that the adjustment of \ gamma does not contribute much to it.

- While negative \gamma values seem to be important from the experimental results, Section 3.3 points out that they may not be strictly proper (scoring rule). If so, how does a negative \gamma affect theoretical results derived in this paper?

- I think it would be better to explain the key idea of convergence analysis based on biased gradient estimator in the main text.

**Time Spent Reviewing:**

20h

---

> ### Author Response · Authors · 2021-08-10
> **Author Response**
>
> Thanks for your valuable feedback! Besides the general response with new experimental results and discussions, we have addressed your questions below:
> 1) How much of the effect of the model-misspecification can be absorbed by adjusting the $\gamma$?
>
> In case of model-misspecification, the true parameter is not contained in the parameter space, so in principle we cannot solve the issue (i.e. finding a solution that perfectly fits the data) by choosing a suitable proper scoring rule. However, a promising strategy is to use various scoring rules (instead of a single one) when finding the solutions in the parameter space to achieve different tradeoffs, which may be important for different downstream tasks.
>
> Consider the following example: the true parameter is $\theta^* = [1,1,1]$, the parameter space is $\Theta = ([0,0,2], [1,1,3])$ and we use $\ell_p$-norm to find the solution $\min_{\theta \in \Theta} \|\theta - \theta^*\|_p$. When we use $\ell_1$-norm, we will get $[1,1,3]$ as the solution and when we use $\ell_2$-norm, we will get $[0,0,2]$. Depending on the specific applications, we may prefer the first or second one, while having a family of proper loss allows us to flexibly choose the solution accordingly. As a generalization of CD, our PS-CD framework is able to provide such modeling flexibility without additional computational cost. Besides generative modeling, in the experiments in general response, we also discovered another valuable property of PS-CD, robustness to data contamination, where different $\gamma$ values can also help achieve different levels of robustness.
>
> 2)  How does a negative $\gamma$ affect theoretical results derived in this paper?
>
> The theoretical analysis on sample complexity and convergence rate also holds when $\gamma < 0$. Since our empirical investigation shows promising results for negative $\gamma$, we also established its connection to Renyi entropy regularized maximum likelihood in Lemma 2.
>
> 3) It would be better to explain the key idea of convergence analysis based on biased gradient estimator in the main text:
> Thanks for the suggestion. We will revise our paper accordingly.

---

> > ### Comment · Reviewer_CEzj · 2021-09-02
> > **thanks to the authors' response**
> >
> > Thank you for responding to the review comments and presenting additional experimental results.
> > In particular, the result that robustness against data contamination can be guaranteed is very interesting and further reinforces the effectiveness of the proposed method. Although I will not change my score, I recommend again to accept.
> >
> > What I found interesting about the added experimental results (it doesn't affect the score, but it would be interesting to consider the reasons)
> > - It is interesting that in the PS-CD method with \gamma = 0.5, when the contamination ratio was increased from 0.1 to 0.2, the KL value increased sharply (a similar phenomenon was also observed when the contamination ratio was increased from 0.2 to 0.3 at \gamma = 1.0).
> >
> > - Compared to the difference between the OOD detection results of the CD and PS-CD methods presented, the difference in the experimental results of robustness between the two methods seems to be quite significant (both in experiments with  artificial data and with MNIST and CIFAR datasets).

---

> > > ### Author Response · Authors · 2021-09-02
> > > **Thanks a lot for your support!**
> > >
> > > Thanks for the feedback and we really appreciate your support for our work!
> > >
> > > About the observations on the added experimental results:
> > > - Yes, in the synthetic data (and real data) experiments, we found that traditional CD/MLE method is vulnerable to even a small amount of data contamination (e.g. a ratio of 0.1 will cause the model to diverge), while by contrast our PS-CD framework shows much better robustness to data contamination. Moreover, for a certain $\gamma > 0$, the model is very robust until the contamination ratio is larger than a threshold; also the contamination ratio threshold for a larger $\gamma$ is also larger, which is in line with the theoretical analysis on the role of $\gamma$ for robust parameter estimation. We will investigate more into this and provide detailed discussions in the updated version.
> > > - Yes, basically we observe that models trained by both CD and PS-CD on **clean training data** have a reasonable ability of out-of-distribution detection. However, when the **training data is contaminated** (e.g. with uniform noise), PS-CD naturally has a mechanism during gradient updates to mitigate the negative effects from the outliers (i.e., the weight before the energy gradient will be close to zero), while CD (although being able to detect the outliers) cannot remove their negative effects and simply treats them equally as clean data during gradient updates, which will cause the model to diverge after a few steps.

---

### Author Response · Authors · 2021-08-10
**Author Response with New Experimental Results**

We thank all the reviewers for providing constructive feedback. We are glad that all reviewers appreciate the novelty, clarity and significance of our work (e.g. “The presented work is considered to have a high impact”, “The theoretical derivation of a novel EBM loss is quite original and the generalizations of CD might influence future practitioners”, “These contributions are solid, and the accompanying theory & experiments are sufficient to convince me”, “The idea is relatively novel and the paper is clearly written”).

We have carefully addressed your comments and inspired by the reviewers’ suggestions, we have conducted new experiments on out-of-distribution (OOD) detection and robust EBM learning under data contamination. We believe that these results can further demonstrate the practical benefits brought by our PS-CD framework and will significantly strengthen our paper.

### OOD Detection
First, we investigate the OOD detection ability of PS-CD. For our conditional CIFAR-10 model, we follow the evaluation protocol in [1] and use $s(x) = \max_y -E(x,y)$ as the score for detecting outliers (details can be found in the accompanying code of [1]). We use SVHN, DTD Textures, Uniform/Gaussian Noise, CIFAR-10 Linear Interpolation and CelebA as the OOD datasets. The results (AUROC score) is summarized in the following table:

| OOD Dataset            | PixelCNN++ | Glow |  CD  | PS-CD |
|------------------------|:----------:|:----:|:----:|:-----:|
| SVHN                   |       0.32 | 0.24 | 0.43 |  0.56 |
| Textures               |       0.33 | 0.27 | 0.36 |  0.44 |
| Uniform/Gaussian Noise |          1 |    1 |    1 |     1 |
| CIFAR-10 Interpolation |       0.71 | 0.59 | 0.63 |  0.68 |
| CelebA                 |          \ |    \ | 0.51 |  0.58 |

From the table we can see that the PS-CD consistently outperforms the traditional CD model in OOD detection tasks.

### Robustness to Data Contamination
Inspired by the OOD detection performance and previous work on robust parameter estimation under data contamination [2], we further test the robustness of CD and PS-CD on both synthetic data and natural image datasets. Specifically, suppose $p(x)$ is the data distribution that we want to approximate with a parametric distribution $q_\theta(x)$, and there is another contamination distribution $\omega(x)$, e.g. uniform noise. In generative modeling under data contamination, our model $q_\theta(x)$ observes i.i.d. samples **from the contaminated distribution** $\tilde{p}(x) = c p(x) + (1 - c) \omega(x)$, where $1 - c \in [0,1/2)$ is the contamination ratio. A theoretical advantage of PS score is its robustness to data contamination: the optimal solution of $\min_\theta S_{sph}(\tilde{p}, q_\theta)$ is close to that of $\min_\theta S_{sph}(p, q_\theta)$ under some conditions (e.g. $\langle \omega q_\theta^\gamma \rangle$ is small around $\theta=\theta^*$, i.e. the density of $\omega(x)$ mostly lies in the region for which target density $p(x)$ is small) [2,3].

In the following, we will show that our practical PS-CD framework enables robust and stable EBM learning under data contamination in both synthetic data and natural image modeling, while traditional CD is unstable and diverge after training with contaminated data.

- Synthetic data: we use $\mathcal{N}(-1, 0.5)$ as the target distribution $p(x)$ and $\mathcal{N}(2, 0.05)$ as the contamination distribution. We train the model distribution $\mathcal{N}(\mu_\theta, \sigma_\theta^2)$ (initialized as a standard Normal) using CD and PS-CD under different contamination ratios and we measure the KL divergence between target distribution $p$ and converged model distribution $q_\theta$, $\mathrm{KL}(p || q_\theta)$, in the following table:

| Contamination Ratio |   CD   | PS-CD ($\gamma = 0.5$) | PS-CD ($\gamma=1.0$) | PS-CD ($\gamma=2.0$) |
|:-------------------:|:------:|:----------------------:|:--------------------:|:--------------------:|
|         0.01        | 0.0067 |                0.00001 |             1.00E-07 |             1.00E-06 |
|         0.05        | 0.0851 |                0.00027 |             1.60E-06 |              0.00011 |
|         0.1         | 0.1979 |                0.00173 |             1.86E-06 |              0.00012 |
|         0.2         | 0.3869 |                 0.1858 |             6.40E-06 |              0.00017 |
|         0.3         | 0.5438 |                 0.5429 |               0.3118 |              0.00029 |

From the above table we can see that the CD algorithm suffers from data contamination severely: as the contamination ratio increases, the performance degrades drastically. By contrast, PS-CD shows good robustness against data contamination and a larger $\gamma$ leads to better robustness. For example, PS-CD with $\gamma=1.0$ can properly approximate the target distribution when the contamination ratio is 0.2, while PS-CD with $\gamma=2.0$ can do so when the contamination ratio is up to 0.3.

- MNIST and CIFAR-10: To show the practical advantage of PS-CD in face of data contamination, we further conduct experiments on MNIST and CIFAR-10 datasets, where we use random uniform noise as the contamination distribution and the contamination ratio is 0.1 (i.e. 10% images in the training set are replaced with random noise). After a warm-up pretraining (when the model has some OOD detection ability), we train the model with the contaminated data and measure the training progress of CD and PS-CD. Note that it is impossible for a randomly initialized model to be robust to data contamination since without additional inductive bias, it will simply treat the contaminated distribution $\tilde{p}$ as the target.

The samples after training with contaminated data can be found here:
- MNIST: https://pasteboard.co/KfeuFZW.png
- CIFAR-10: https://pasteboard.co/KfevqvT.png

We can see that CD gradually generates more random noise and diverge after a few training steps, while
PS-CD is very robust. In particular, for a slightly pretrained unconditional CIFAR-10 model (simple 5-layer CNN with FID of 68.77), we observe that the performance of CD degrades drastically in terms of FID, while PS-CD can continuously improve the model even using the contaminated data:

|     | Pretrained Model | CD 1000 steps | CD 2000 steps | PS-CD 1000 steps | PS-CD 2000 steps |
|-----|:----------------:|:-------------:|:-------------:|:----------------:|:----------------:|
| FID |            68.77 |         95.56 |        300.89 |            59.78 |            57.24 |

- Interpretation: We believe that robustness to data contamination is a valuable property for modern deep generative models and there is actually a natural interpretation for the robustness of PS-CD. Compared to CD, there is an extra weight term before the gradient of the energy: $\frac{\exp(- \gamma E_\theta(x_i))}{\sum_j \exp(- \gamma E_\theta(x_j))} \nabla_\theta E_\theta(x_i)$ (the first term in Eq. (19)). Suppose $x_i \sim \omega$ is a noise data in a batch of samples from the contaminated distribution $\tilde{p}$, for a model with OOD detection ability, it will assign a much higher energy to $x_i$ than normal data and the weight before $\nabla_\theta E_\theta(x_i)$ will be close to zero. In short, PS-CD naturally integrates the OOD detection ability of EBMs into the **training process**, which then leads to robustness to data contamination.

We will incorporate these results and discussions in the updated version and we addressed your specific questions below.

[1] Implicit Generation and Modeling with Energy-Based Models

[2] Robust estimation under heavy contamination using unnormalized models

[3] Robust parameter estimation with a small bias against heavy contamination

---

### Decision · Program_Chairs · 2021-09-27

**Decision:**

Accept (Poster)

**Comment:**

This paper introduced a class of divergence to learn energy-based models. The proposed divergence class has an auxiliary loss parameter that can balance the trade-off between diversity and quality. The method was applied to image benchmark datasets. The reviewers agree that this is an interesting paper with nice contributions. We conclude that the paper is worthy of acceptance.